# Representer Point Selection via Local Jacobian Expansion for Post-hoc Classifier Explanation of Deep Neural Networks and Ensemble Models

**Yi Sui**
University of Toronto
yi.sui@mail.utoronto.ca

**Ga Wu**[*]
Borealis AI
ga.wu@borealisai.ca

**Scott Sanner**[†]
University of Toronto
ssanner@mie.utoronto.ca

## Abstract

Explaining the influence of training data on machine learning model predictions is a critical tool for debugging models through data curation. A recent appealing and efficient approach for this task was provided via the concept of Representer Point Selection (RPS), i.e. a method the leverages the dual form of $l_2$ regularized optimization in the last layer of the neural network to identify the contribution of training points to the prediction. However, two key drawbacks of RPS-$l_2$ are that they (i) lead to disagreement between the originally trained network and the RPS-$l_2$ regularized network modification and (ii) often yield a static ranking of training data for test points in the same class, independent of the test point being classified. Inspired by the RPS-$l_2$ approach, we propose an alternative method based on a local Jacobian Taylor expansion (LJE). We empirically compared RPS-LJE with the original RPS-$l_2$ on image classification (with ResNet), text classification recurrent neural networks (with Bi-LSTM), and tabular classification (with XGBoost) tasks. Quantitatively, we show that RPS-LJE slightly outperforms RPS-$l_2$ and other state-of-the-art data explanation methods by up to 3% on a data debugging task. More critically, we qualitatively observe that RPS-LJE provides stable and individualized explanations that are more coherent to each test data point. Overall, RPS-LJE represents a novel approach to RPS-$l_2$ that provides a powerful tool for sample-based model explanation and debugging.

## 1 Introduction

Deep learning as well as ensemble methods such as XGBoost [2] have revolutionized the field of machine learning and led to unprecedented levels of accuracy in a variety of data-driven prediction applications. However, it can be extremely challenging to debug these complex methodologies when they make incorrect predictions. The first step in this debugging process is *explaining* the prediction, which can stem from a variety of interpretive processes. For example, we may seek to understand the key features [13, 16] or salient regions [18, 20] that led to a prediction, or in a more recent line of work, to understand the influence of the training data on test data predictions [1, 11, 15, 23].

In this paper we focus on a particularly appealing (and efficient) method for understanding the impact of training data on test predictions, namely Representer Point Selection (RPS) [23], which leverages an application of the representer theorem [17] to deep neural networks. Specifically, it uses the dual form of $l_2$ regularized optimization in the last layer of the neural network to identify the contribution of training data to the test prediction. Compared to model-agnostic approaches [1, 9, 10] that estimate

---

[*]Contributions were made while the author was at the University of Toronto.
[†]Affiliate to Vector Institute of Artificial Intelligence, Toronto.

35th Conference on Neural Information Processing Systems (NeurIPS 2021).

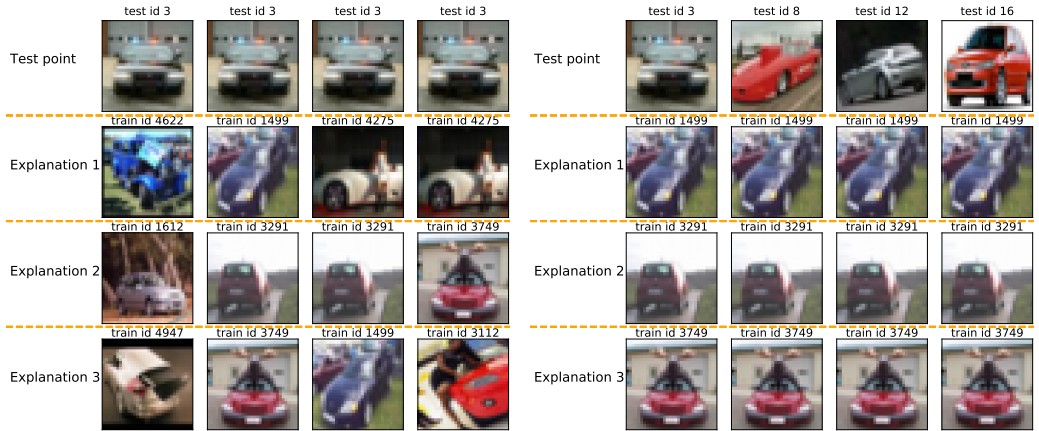

(a) Same test image, different $l_2$ weight.    (b) Different test image, same $l_2$ weight.

Figure 1: **Drawbacks of Current Representer Point Selection Explanation (RPS-$l_2$) on CIFAR-10 Dataset.** The target model is ResNet [6]. (a) Explanations vary when $l_2$ regularized fine-tuning of RPS-$l_2$ is conducted with different hyperparameter weightings. $l_2$ coefficient value of each column from left to right:[1e-5, 1e-4, 1e-2, 1e-1] (b) RPS-$l_2$ produces *identical* explanations (and rankings) for different test samples that belong to the same predicted class.

data influence after optimization, the RPS approach directly integrates with the prediction model to provide a high-fidelity white-box interpretation of the prediction. Compared to the influence function-based approach [11], the RPS is more computationally efficient as it focuses on the last layer of a neural network instead of *all* parameters.

While holding many advantages, we note the existing RPS approach faces two key drawbacks. First, as the RPS approach requires fine-tuning the last layer of the neural network with $l_2$ regularization, it leads to a disagreement between the originally trained network and the $l_2$ regularized network and is furthermore highly sensitive to $l_2$ regularization strength (cf. Figure 1(a)). Thus, it is hard to justify if the explanation produced by RPS is for the original model or the modified model. Second, we observe that the explanations produced by RPS are nearly identical for all test samples that are classified into the same category, which provides more of a class-level rather than instance-level explanation for RPS (cf. Figure 1(b)). While the RPS approach is appealing in principle, both of these drawbacks significantly harm the utility of the RPS explanation approach in practice.

To address these issues, this paper presents an alternative derivation for Representer Point Selection (RPS) based on a Local Jacobian Taylor expansion (LJE), which corrects for both aforementioned problems. We empirically compared RPS-LJE to the RPS-$l_2$ on image classification (with ResNet [6]), text classification (with Bi-LSTM), and credit analysis (with XGBoost [2]) tasks. Quantitatively, we show RPS-LJE outperforms RPS-$l_2$ and other state-of-the-art data explanation methods by to up 3% on a data debugging task. Qualitatively — and perhaps most importantly — the RPS-LJE provides stable (i.e., no need for $l_2$ tuning) and diverse explanations that are more coherent to the test data.

## 2 Representer Points Selection for Explaining Deep Neural Networks

### 2.1 Preliminaries

In a machine learning context, representer theorems [17] loosely state that under certain conditions, model prediction $\hat{\mathbf{y}}_t$ of a test sample $x_t$ can be expressed as a linear combination of kernel evaluations $\mathcal{K}(\mathbf{x}_i, \mathbf{x}_t)$ between each training point $\mathbf{x}_i$ and the test sample $\mathbf{x}_t$ such that

$$\hat{\mathbf{y}}_t = \sum_i^n \boldsymbol{\alpha}_i \mathcal{K}(\mathbf{x}_i, \mathbf{x}_t) \, , \tag{1}$$

where $\boldsymbol{\alpha}_i$ is the weight of the training data point $i$ that is independent from the test point $t$.

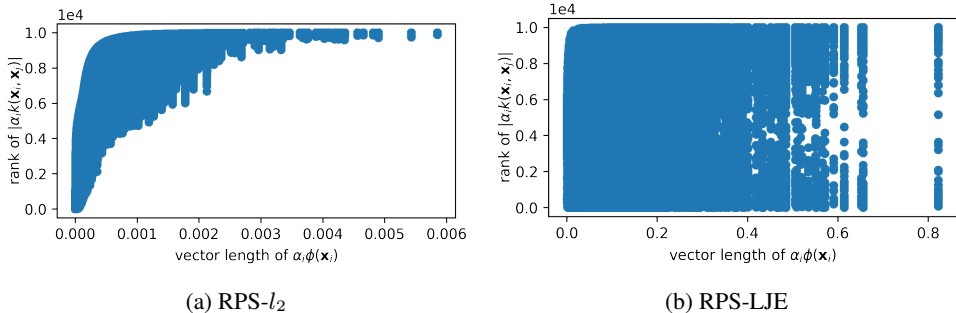

(a) RPS-$l_2$                    (b) RPS-LJE

Figure 2: **Correlation between the Rank of Explanations $(\alpha_i\phi(\mathbf{x}_i)^T\phi(\mathbf{x}_t))$ and term $\alpha_i\phi(\mathbf{x}_i)$'.**
We show results for a binary classification task (horses vs. cars), where we only look at $\alpha_i$ of the positive class. (a) RPS-$l_2$'s explanation rank heavily depends on $\alpha_i\phi(\mathbf{x}_i)$, which shows strong dominance of $\alpha_i\phi(\mathbf{x}_i)$ that suppresses information in the test example $\mathbf{x}_t$ being explained. (b) The proposed RPS-LJE explanation is not dominated by $\alpha_i\phi(\mathbf{x}_i)$, which instead varies widely with $\mathbf{x}_t$.

As the representer theorem linearly separates the contribution of training samples to the prediction, it has been introduced into the deep learning model interpretation research [23] for tracking the training data influence on predictions. Concretely, the current Representer Point Selection (RPS) approach [23] introduces an $l_2$ regularization term (for the last layer's parameters $\Theta_L \subset \boldsymbol{\Theta}$) into a model's objective function, which facilities its derivation such that it fulfills Equation 1 by setting

$$\boldsymbol{\alpha}_i = -\frac{1}{2\lambda n}\frac{\partial\mathcal{L}(\mathbf{x}_i, y_i, \boldsymbol{\Theta})}{\partial\Theta_L\phi(\mathbf{x}_i)} \quad and \quad \mathcal{K}(\mathbf{x}_i, \mathbf{x}_t) = \phi(\mathbf{x}_i)^T\phi(\mathbf{x}_t)\,, \tag{2}$$

where $\lambda$ denotes the hyper-parameter of $l_2$ regularization term, $n$ denotes the number of total training samples, and $\phi(\mathbf{x}_i)$ and $\phi(\mathbf{x}_t)$ are the representations of input $\mathbf{x}_i$ and $\mathbf{x}_t$ at layer $L$. As this explanation approach works on the last layer of a network and is efficient to compute, it demonstrates a significant advantage over Influence Function-based approaches [11] in terms of computational efficiency.

## 2.2 Caveats of Current Representer Points Selection

As mentioned, to facilitate its derivation, the current RPS approach introduces an $l_2$ regularization term into the computation, which inevitably violates the saddle point assumption its derivation relies on. To address this, the RPS conducts fine-tuning on the trained model with a new objective function

$$\Theta_L^* = \underset{\Theta_L}{\arg\min}\left\{\frac{1}{n}\sum_i^n\mathcal{L}(\Phi(x_i, \boldsymbol{\Theta}), \Phi(x_i, \boldsymbol{\Theta}_{given})) + \lambda\|\Theta_L\|^2\right\}\,, \tag{3}$$

where $\Phi(x_i, \boldsymbol{\Theta}_{given})$ and $\Phi(x_i, \boldsymbol{\Theta})$ represents the post-activation prediction of $x_i$ of the given model and fine-tuned model, respectively. Correspondingly, the kernel weights in the expression of Equation 2 are updated as

$$\boldsymbol{\alpha}_i = -\frac{1}{2\lambda n}\frac{\partial\mathcal{L}(\mathbf{x}_i, y_i, \boldsymbol{\Theta}^*)}{\partial\Theta_L^*\phi(\mathbf{x}_i)}. \tag{4}$$

As the data importance factor $\boldsymbol{\alpha}_i$ is computed with respect to the $l_2$-modified parameter set $\boldsymbol{\Theta}^*$, the explanations correspond to the modified model instead of the original model. Indeed, the gap between the two sets of parameters is sensitive to the hyper-parameter setting of the $l_2$ regularizer. This reflects our previous observation in Figure 1(a) showing that prediction explanations can vary *significantly* based on different $l_2$ regularizer weightings.

Also, by inspecting the expression of training data contribution

$$\Theta_L^*\phi(\mathbf{x}_t) = \boldsymbol{\alpha}_i\mathcal{K}(\mathbf{x}_i, \mathbf{x}_t) = -\frac{1}{2\lambda n}\frac{\partial\mathcal{L}(\mathbf{x}_i, y_i, \boldsymbol{\Theta}^*)}{\partial\Theta_L^*\phi(\mathbf{x}_i)}\phi(\mathbf{x}_i)^T\phi(\mathbf{x}_t)$$

$$= -\frac{1}{2\lambda n}\frac{\partial\mathcal{L}(\mathbf{x}_i, y_i, \boldsymbol{\Theta}^*)}{\partial\Theta_L^*\phi(\mathbf{x}_i)}\frac{\partial\Theta_L^*\phi(\mathbf{x}_i)}{\partial\Theta_L^*}\phi(\mathbf{x}_t) = \underbrace{-\frac{1}{2\lambda n}\frac{\partial\mathcal{L}(\mathbf{x}_i, y_i, \boldsymbol{\Theta}^*)}{\partial\Theta_L^*}}_{dominant\ term}\phi(\mathbf{x}_t), \tag{5}$$

we note the first-order derivative of the loss of a single training example is the dominant term that suppresses a small difference of test example representation $\phi(\mathbf{x}_t)$ among test samples in the same class. Figure 2(a) shows the strong positive correlation between the dominant term and the relevant order of the training data contribution. This supports our observation in Figure 1(b) that explanations tend to be identical for the test samples being classified into the same category making RPS more of a class-level than instance-level explanation that defeats the intent of uncovering instance-level prediction errors!

## 3   Representer Point Selection via Local Jacobian Expansion

In an effort to preserve the conceptual and computational advantages of the previously discussed RPS-$l_2$ methodology while improving fidelity of the explanations to the original model and encouraging test instance-level explanation, we now present a novel derivation of Representer Point Selection through a Local Jacobian Expansion (RPS-LJE). Similar to RPS-$l_2$, RPS-LJE also expresses the pre-activation prediction outcome of a well-trained classification model (near the saddle point) as a linear combination of kernel evaluations between the test sample and the training points. Specifically, we use a first-order Taylor expansion on the Jacobian matrix in our derivation, which avoids the problems introduced by imposing the additional $l_2$ regularization term in RPS-$l_2$. First, we begin with the formal problem definition.

Consider a classifier $\mathcal{M}_{\boldsymbol{\Theta}^\dagger}$ (the target model in the following context) that has learned to map input observation $\mathbf{x}_t \in \mathbb{R}^d$ to an output space $y_t \in \{1 \cdots k\}$, whose pre-activation prediction[3] $\bar{\mathcal{M}}_{\boldsymbol{\Theta}^\dagger}$ could be formulated as

$$\hat{\mathbf{y}}_t = \bar{\mathcal{M}}_{\boldsymbol{\Theta}^\dagger}(\mathbf{x}_t) = \Theta_L^\dagger \phi(\mathbf{x}_t), \tag{6}$$

where bold $\mathbf{y}_i \in \mathbb{R}^k$ represents the prediction vector (one element for a class). Our goal of prediction explanation is to reformulate Equation 6 into the format of Equation 1 such that the contribution of each training data point on the prediction is linearly separable.

While the above setting appears restricted, it represents a large group of machine learning models commonly used in practice; many well-known models, such as ResNet [6] for images, Transformers [22] for text, and even XGBoost [2] for tabular classification tasks, can be expressed in this simple formulation leveraging a feature embedding stage followed by a pre-activation linear prediction stage.

### 3.1   First-order Taylor Expansion on Jacobian Matrix

We begin by presuming we are given a well-trained target model $\mathcal{M}_{\boldsymbol{\Theta}^\dagger}$, whose loss derivative with respect to the decision making parameter $\Theta_L$ is close to a saddle point such that

$$0 \approx \sum_{i=1}^{n} \left.\frac{\partial \mathcal{L}(\mathbf{x}_i, y_i, \boldsymbol{\Theta})}{\partial \Theta_L}\right|_{\Theta_L = \Theta_L^\dagger}. \tag{7}$$

Here, we assume that the loss term $\mathcal{L}$ is twice-differentiable and strictly convex with respect to the last linear layer parameters $\Theta_L$. In the case of non-convexity, we can adopt the approach proposed in the Influence Function [11] to form a convex quadratic approximation of the loss by introducing a damping term .

With any parameter assignment $\Theta_L^*$ close to the model parameter $\Theta_L^\dagger$, we can further rewrite Equation 7 via a first-order Taylor expansion such that

$$0 \approx \frac{1}{n} \sum_{i=1}^{n} \underbrace{\left.\frac{\partial \mathcal{L}(\mathbf{x}_i, y_i, \boldsymbol{\Theta})}{\partial \Theta_L}\right|_{\Theta_L = \Theta_L^*}}_{\nabla_{\Theta_L} \mathcal{L}(\mathbf{x}_i, y_i, \boldsymbol{\Theta}^*)} + (\Theta_L^\dagger - \Theta_L^*) \underbrace{\left[\frac{1}{n} \sum_{i=1}^{n} \left.\frac{\partial \mathcal{L}^2(\mathbf{x}_i, y_i, \boldsymbol{\Theta})}{\partial \Theta_L^2}\right|_{\Theta_L = \Theta_L^*}\right]}_{\mathcal{H}_{\Theta_L^*} \overset{\text{def}}{=} \frac{1}{n} \sum_{i=1}^{n} \nabla_{\Theta_L}^2 \mathcal{L}(\mathbf{x}_i, y_i, \boldsymbol{\Theta}^*)}. \tag{8}$$

This expansion yields the model parameter $\Theta_L^\dagger$ as a linear combination of functions of each training data point

$$\Theta_L^\dagger = \Theta_L^* - \sum_{i=1}^{n} \frac{1}{n} \mathcal{H}_{\Theta_L^*}^{-1} \nabla_{\Theta_L} \mathcal{L}(\mathbf{x}_i, y_i, \boldsymbol{\Theta}^*) + \xi, \tag{9}$$

---

[3] For the trained classifier, the predictions before and after activation are consistent.

where $\xi$ is a negligible error term. Intuitively, this equation shows the model parameter $\Theta_L^\dagger$ could be reconstructed through a one-step gradient descent from the nearby parameter assignment $\Theta_L^*$ with a dynamic learning rate $\frac{1}{n}\mathcal{H}_{\Theta_L^*}^{-1}$ (as a matrix).

We remark that the expression in Equation 9 represents the contribution of each data point as a linearly separable function, which is fundamentally different from recording gradients at training time as a data importance score [15]. That is, the gradients during model training are sensitive to both training order and optimizer settings.

For Equation 9 to hold after the Taylor expansion, $\Theta_L^*$ has to be close to the model parameter $\Theta_L^\dagger$. Therefore, we propose to estimate $\Theta_L^*$ through a one-step stochastic gradient ascent from the trained model using any optimizer (Adam, RMSProp, etc), such that

$$\sum_{i=1}^n \mathcal{L}(\mathbf{x}_i, y_i, \Theta^*) > \sum_{i=1}^n \mathcal{L}(\mathbf{x}_i, y_i, \Theta^\dagger) \tag{10}$$

and $\Theta_L^*$ is maintained close to the original model parameter $\Theta_L^\dagger$ with a small loss shift.

## 3.2 Representer Point Selection with Local Jacobian Expansion (RPS-LJE)

With the derivation of Equation 9, we can now reformulate the pre-activation prediction of a test point $\mathbf{x}_t$ as a weighted linear combination of kernels leading to our key result of the final form of RPS-LJE (see Appendix B for a more detailed derivation):

$$\begin{aligned}
\Theta_L^\dagger \phi(\mathbf{x}_t) &= \sum_{i=1}^n \left[ \frac{1}{n}\Theta_L^* - \frac{1}{n}\mathcal{H}_{\Theta_L^*}^{-1} \nabla_{\Theta_L} \mathcal{L}(\mathbf{x}_i, y_i, \Theta^*) \right] \phi(\mathbf{x}_t) \\
&= \sum_{i=1}^n \underbrace{\left[ \Theta_L^* \frac{1}{\phi(\mathbf{x}_i)n} - \frac{1}{n}\mathcal{H}_{\Theta_L^*}^{-1} \frac{\partial \mathcal{L}(\mathbf{x}_i, y_i, \Theta^*)}{\partial \Theta_L^* \phi(\mathbf{x}_i)} \right]}_{\boldsymbol{\alpha}_i} \underbrace{\phi(\mathbf{x}_i)^T \phi(\mathbf{x}_t)}_{\mathcal{K}(\mathbf{x}_i, \mathbf{x}_t)} .
\end{aligned} \tag{11}$$

Similar to the claim of the original RPS using an $l_2$ norm (RPS-$l_2$), when a training data $\mathbf{x}_i$ is close to the test point $\mathbf{x}_t$ in the representation space with a large positive value $\alpha_{ik}$, the prediction score for class $k$ is increased. On the other hand, when the $\alpha_{ik}$ is a large negative value, the prediction score for class $k$ is then decreased.

The first notable difference between our derivation of RPS-LJE and the original RPS-$l_2$ is that the $\boldsymbol{\alpha}_i$ term now contains an inverse of second-order derivative that estimates the correlation among the parameter entries in $\Theta_L^\dagger$. This modification mitigates the risk of over-weighting training data points with a large predictive error that causes a small number of data points to dominate the explanations as discussed in Section 2.2 and Figure 1(b). Figure 2(b) illustrates the effects of correction.

The second difference is that the prediction explanation of our derivation of RPS-LJE is faithful w.r.t. the original model $\mathcal{M}_{\Theta^\dagger}$ instead of the $l_2$-regularized model $\mathcal{M}_{\Theta^*}$. To clarify, the left hand side of of Equation 11 is different with that of RPS-$l_2$ in Equation 5. In addition, data importance factor $\boldsymbol{\alpha}_i$ in the new derivation no longer depends on the $l_2$ regularization hyper-parameter. Hence, the changes in the RPS-LJE framework directly address the problem described in Figure 1(a).

## 3.3 Relation to Influence Function-based Interpretation

The Influence Function method [11] estimates the prediction importance of each training data point by up-weighting the data points with small perturbation $\epsilon$ that results in the following final expression

$$\mathcal{I}_{up,loss}(\mathbf{x}_i, \mathbf{x}_t) = -\nabla_{\Theta_L} \mathcal{L}(\mathbf{x}_t, y_t, \Theta^\dagger)^T \underbrace{\mathcal{H}_{\Theta_L^\dagger}^{-1} \nabla_{\Theta_L} \mathcal{L}(\mathbf{x}_i, y_i, \Theta^\dagger)}_{\mathcal{I}_{up,params}(\mathbf{x}_i) \stackrel{\text{def}}{=} \frac{d\mathcal{L}(\mathbf{x}_t, y_t, \Theta^\dagger)}{d\epsilon}\big|_{\epsilon=0}} . \tag{12}$$

If we rewrite the above equation by expanding the first-order derivatives with chain rule such that

$$\mathcal{I}_{up,loss}(\mathbf{x}_i, \mathbf{x}_t) = -\frac{\partial \mathcal{L}(\mathbf{x}_t, y_t, \Theta^\dagger)}{\partial \Theta_L^\dagger \phi(\mathbf{x}_t)} \mathcal{H}_{\Theta_L^\dagger}^{-1} \frac{\partial \mathcal{L}(\mathbf{x}_i, y_i, \Theta^\dagger)}{\partial \Theta_L^\dagger \phi(\mathbf{x}_i)} \phi(\mathbf{x}_i)^T \phi(\mathbf{x}_t), \tag{13}$$

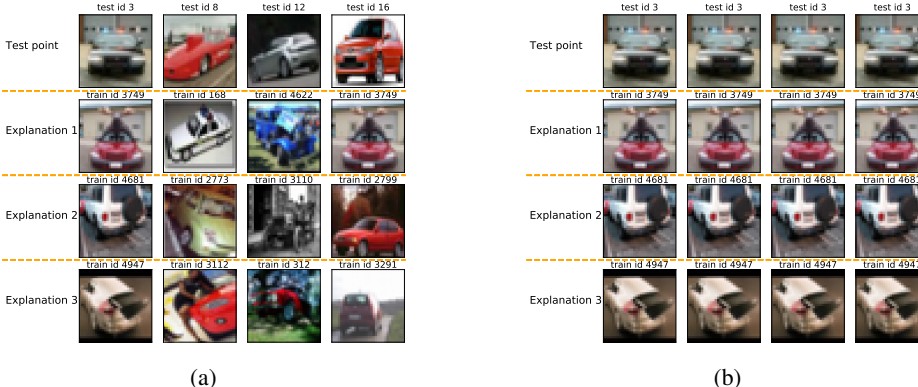

(a)                                        (b)

Figure 3: **Sanity Check of Representer Point Selection with Local Jacobian Expansion (RPS-LJE) on CIFAR-10 Dataset.** The target model is ResNet [6] (a) RPS-LJE can correctly produce individualized explanations. (b) Explanation maintains consistency as the gradient hyperparameter setting changes. Learning rate of each column from left to right: $[5e-4, 1e-3, 5e-3, 1e-2]$.

we note the expression is very similar to RPS-LJE proposed in this work (as described in Equation 11) except two subtle differences:

- The Influence Function includes a multiplicative factor $\frac{\partial \mathcal{L}(\mathbf{x}_t, y_t, \Theta^{\dagger})}{\partial \Theta_L^{\dagger} \phi(\mathbf{x}_t)}$ that is relevant to the test point, whereas the RPS-LJE has an additive factor $\frac{1}{n} \Theta_L^* \phi(\mathbf{x}_t)$ that is also only relevant to the test point.
- The derivative in the Influence Function respects the original model parameters $\Theta^{\dagger}$, whereas RPS-LJE's derivatives apply to $\Theta^*$.

Ultimately, these differences help explain the comparative performance of these methods in our experiments (cf. Section 4.3), but — because they are also subtle — they help explain the similarity of explanations produced by RPS-LJE and Influence Functions for image classification explanation.

## 4 Experimental Evaluation

We perform a range of experiments with multiple datasets (and corresponding model architectures) and evaluate the performance of RPS-LJE against the original RPS-$l_2$ as well as Influence Function-based approaches. Note that all three methods do not specifically require access to the training history, which is rarely available in deployment settings. In contrast, TracIn [15], which requires access to training checkpoints, is not in the scope of our comparison. The goal of these experiments is to demonstrate that the alternative derivation of RPS described in this work, RPS-LJE, successfully addresses the two critical drawbacks of RPS-$l_2$ and leads to substantial performance improvement on multiple use cases, including data debugging and model behavior explanation. All code to reproduce these results is publicly available on Github.[4]

### 4.1 Sanity Check of Representer Point Selection with Local Jacobian Expansion

Before describing our quantitative analysis, we first start with a sanity check to show the proposed RPS-LJE approach indeed addresses the problem we highlighted earlier in Section 2.2 and Figure 1. Concretely, we repeat our showcase in Figure 3 using RPS-LJE with the same test examples and target model. Here, we highlight the following observations:

- As shown in Figure 3(a), the RPS-LJE can produce an individualized explanation for test samples in the same category. This observation reflects our previous description in Figure 2(b), where the term $\boldsymbol{\alpha}_i \phi(\mathbf{x}_i)$ in RPS-LJE no longer dominates the explanation ranking.

---

[4]`https://github.com/echoyi/RPS_LJE`

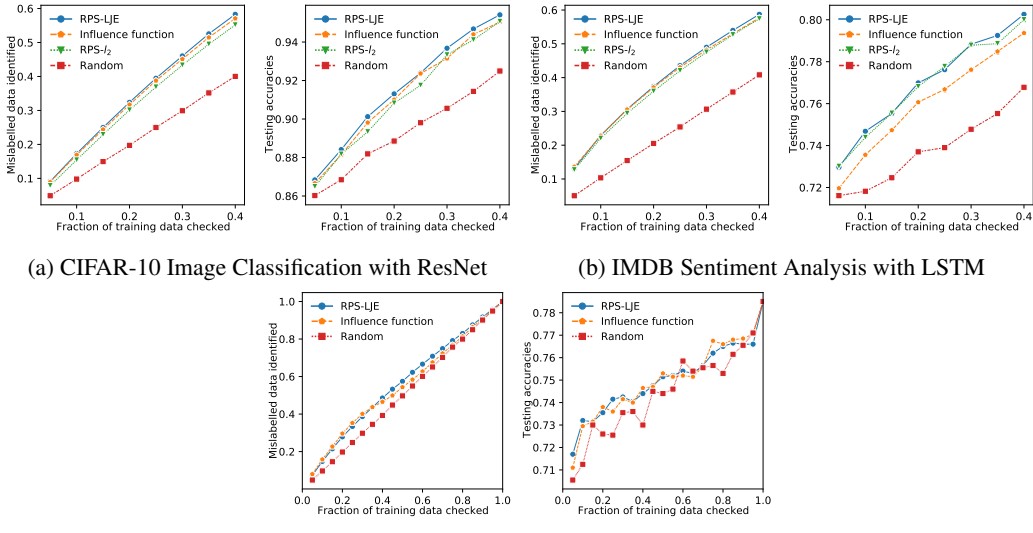

(a) CIFAR-10 Image Classification with ResNet    (b) IMDB Sentiment Analysis with LSTM

(c) German Credit Risk Analysis with XGBoost

Figure 4: **Performance Comparison on Dataset Debugging Tasks.** For each sub-figure group, we show data correction rate vs. fraction of data checked (left) and how such a correction would impact performance (right). RPS-LJE shows slightly better performance than Influence Functions and marginally better than the original RPS with $l_2$ norm injection.

- To test the sensitivity of one-step gradient descent in RPS-LJE (used to search $\Theta_L^*$ as described in Section 3.1), we experimented with multiple learning rates $\gamma$ in a large range $\gamma \in \{5e\text{-}4 \cdots 1e\text{-}3, 5e\text{-}3 \cdots 1e\text{-}2\}$. As shown in Figure 3(b), the explanations vary little with the learning rate.

## 4.2 Fixing Mislabeled Training Examples (Data Debugging)

In this experiment, we simulate an application scenario where human experts need to inspect the data annotation quality of the training set that directly impacts the model performance on the test samples.

We simulate data debugging on three classification tasks, including 1) binary image classification with ResNet [6] (ResNet-20) on CIFAR-10 [12] dataset (horses vs. cars) presented in the RPS-$l_2$ paper [23], 2) sentiment analysis with Bi-LSTM [7] on IMDB [14] dataset, and 3) credit risk identification with XGBoost on German Credit dataset [4]. The datasets are intentionally corrupted by randomly flipping 20-30 percent of the data points' labels, naturally resulting in low test accuracy. Our goal is to identify which data points' label corruption are more harmful and correct them as early as possible. With the partially corrected dataset (after each 5% of checking), we retrain the models and record the test accuracies for each task. Experiments are repeated for ten random split and corruptions on CNN and XGBoost and five random split and corruptions on RNN; we report the average result. For RPS approaches (LJE and $l_2$), we pick data points that have the largest self-prediction contribution $|\alpha_i \mathcal{K}(\mathbf{x}_i, \mathbf{x}_i)|$ [5] as suspicious corrupted data points. For Influence Functions, we use self-influence (see Equation 12) as the score of ranking. For the random baseline, we picked the data to check randomly.

Figure 4 shows the experimental results. Here, we highlight the following observations:

- RPS-LJE either slightly dominates RPS-$l_2$ or performs comparably to it. This demonstrates that it is an effective data debugging tool comparable to as RPS-$l_2$, since both show a significant performance gap to the random baseline on all three tasks.
- RPS-LJE shows slightly better performance than both Influence Function and RPS-$l_2$ on all tests. For the image classification task (Figure 4(a)), the RPS-LJE shows 3% better performance (around 60 more mislabelled data identified) than original RPS-$l_2$ after searching through 40% of all training data. For the sentiment analysis task (Figure 4(b)), the performance improvement is about 1% (around 35 more mislabelled data found than the others).

---

[5]We note that, for RPS-$l_2$, there is no difference on using $|\alpha_i|$ or $|\alpha_i \mathcal{K}(\mathbf{x}_i, \mathbf{x}_i)|$ to select corrupt data points.

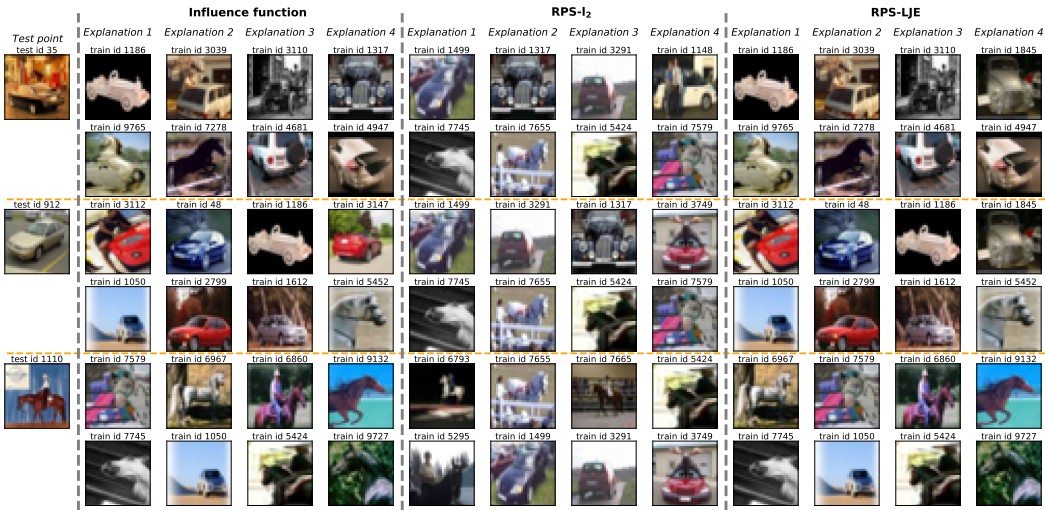

Figure 5: **Comparison of Top-4 Influential Training Images for Three Test Samples.** For each test sample, the upper row shows positive influential points, whereas the bottom row shows negative influential points. Examples are randomly selected from 2000 test samples in our experiment setting.

## 4.3 Understand Model Behavior through Prediction Explanation

In this section, we qualitatively analyze the prediction explanation ability of candidate sample-based explanation models on the three tasks mentioned in Section 4.2. While we compare the explanation results in this experiment, we only intend to compare and contrast the three explanation approaches.

### CIFAR-10 Image Classification with ResNet

In Figure 5, we visualize the top-4 training points (both positively and negatively) that have the strongest influence on the prediction of test samples from Influence Function, RPS-$l_2$, and the proposed RPS-LJE approaches on the image classification model. Here, we make the following key observations:

- The explanation produced by RPS-LJE is more similar to that of Influence Function than RPS-$l_2$ but with some differences in explanation order. This observation aligns with our previous conjecture in Section 3.3 that the RPS-LJE's formula of computing training data importance appears similar to that of Influence Functions.
- Explanations produced by RPS-$l_2$ contain a lot of repetition. Training example ID-3291 and ID-7655 are in all three test samples' explanation lists.

### IMDB Sentiment Analysis with LSTM

Table 1 shows the top-3 explanations produced by the candidate explanation approaches on the IMDB sentiment analysis model (Bi-LSTM). Here, we make the following key observations:

- Among the three explanation approaches, RPS-LJE's explanation is more coherent to the properties of the test points. For instance, the explanations of RPS-LJE for Sample 1 all start with "This", which hints to the auditor that the model has made a generalization (whether correct or not) that positive sentiment sentences start with "This" based on the identified training data. Similarly, for Sample 1 and 3, we see the explanations follow the same narrative style by starting with "I".
- Explanations provided by Influence Functions generate similar explanations (2 out of the top 3) with RPS-LJE. But Influence Functions sometimes produce confusing explanations. For example, Sample 2 has a positive comment, but Explanation 1 provided by the Influence Function has a negative sentiment. Similarly, Sample 3 and its Explanation 3 have the opposite sentiment.

### German Credit Risk Analysis with XGBoost

Table 2 lists the predictions of three test samples in the German Credit dataset through the XGBoost model with corresponding explanations produced by RPS-LJE and Influence Function. Here, we

Table 1: **Explanation Comparison among RPS-LJE, RPS-$l_2$ and Influence Function on IMDB sentiment analysis data.** Column "Sentiment" is the target (label) column, and raw review is the input of Bi-LSTM network. Examples are randomly selected from the test set.

| | | ID | Sentiment | Raw Reviews |
|---|---|---|---|---|
| **Sample 1** | Test point | 41 | positive | This movie is good for entertainment purposes, but it is not historically reliable. If you are look··· |
| RPS-LJE | Explanation 1 | 741 | positive | This movie is about sexual obsession.Bette Davis plays Mildred. This is a woman who men are··· |
| | Explanation 2 | 14701 | positive | This is a very memorable spaghetti western.It has a great storyline, interesting characters,and some··· |
| | Explanation 3 | 3159 | positive | This movie was featured on a very early episode of Mystery Science Theater 3000,but when I see··· |
| RPS-$l_2$ | Explanation 1 | 9112 | positive | Tim Krabbe is the praised author of 'Het Gouden Ei' , a novel that was put on the screen twice··· |
| | Explanation 2 | 3704 | positive | THE DEVIL'S PLAYTHING is my second attempt at a Joseph Sarno production-and although I ··· |
| | Explanation 3 | 4000 | positive | So , Todd Sheets once stated that he considers his 1993 , shot-on-video Z - epic, Zombie Bloodbath··· |
| Influence function | Explanation 1 | 14701 | positive | This is a very memorable spaghetti western.It has a great storyline, interesting characters,and some··· |
| | Explanation 2 | 741 | positive | This movie is about sexual obsession.Bette Davis plays Mildred. This is a woman who men are··· |
| | Explanation 3 | 669 | positive | Did you ever wonder how far one movie could go?Schizophreniac relentlessly explores the world··· |
| **Sample 2** | Test point | 525 | positive | I can think of no movie that better captures the concept of grace, in a theological sense.The well-··· |
| RPS-LJE | Explanation 1 | 14109 | positive | I can tell you just how bad this movie is.I was in the movie and I haven't seen it yet,but I cringe at··· |
| | Explanation 2 | 2300 | positive | I am one of Jehovah's Witnesses and I also work in an acute care medical facility.Over the years I··· |
| | Explanation 3 | 6372 | positive | I think the film is educational. However,it fails to treat the issue which sparked so much controversy··· |
| RPS-$l_2$ | Explanation 1 | 9112 | positive | Tim Krabbe is the praised author of 'Het Gouden Ei' , a novel that was put on the screen twice··· |
| | Explanation 2 | 3704 | positive | THE DEVIL'S PLAYTHING is my second attempt at a Joseph Sarno production-and although I ··· |
| | Explanation 3 | 4000 | positive | So , Todd Sheets once stated that he considers his 1993 , shot-on-video Z - epic, Zombie Bloodbath··· |
| Influence function | Explanation 1 | 2394 | negative | A memorable line from a short lived show.After viewing the episode where that line was introduced··· |
| | Explanation 2 | 14109 | positive | I can tell you just how bad this movie is.I was in the movie and I haven't seen it yet,but I cringe at··· |
| | Explanation 3 | 2300 | positive | I am one of Jehovah's Witnesses and I also work in an acute care medical facility.Over the years I··· |
| **Sample 3** | Test point | 13087 | negative | I really tried to like this film about a doctor who has the possibility of a new life with a young··· |
| RPS-LJE | Explanation 1 | 3064 | negative | I'm sorry but I didn't like this doc very much I can think of a million ways it could have been better··· |
| | Explanation 2 | 4622 | negative | I have to be completely honest in saying first that I fell asleep somewhere in the middle, so I can··· |
| | Explanation 3 | 9777 | negative | I recently viewed Manufactured Landscapes at the Seattle International Film Festival.I was drawn··· |
| RPS-$l_2$ | Explanation 1 | 4801 | negative | A so common horror story about a luxury building at Brooklyn which hides the gates to hell . It is··· |
| | Explanation 2 | 11015 | negative | The thing that stands out in my mind in this film ( sadly ) is the introduction , where John Berlin··· |
| | Explanation 3 | 12446 | negative | Taped this late night movie when I was in grade 11 , watched it on fast forward . I suggest you do··· |
| Influence function | Explanation 1 | 3064 | negative | I'm sorry but I didn't like this doc very much I can think of a million ways it could have been better··· |
| | Explanation 2 | 4622 | negative | I have to be completely honest in saying first that I fell asleep somewhere in the middle, so I can··· |
| | Explanation 3 | 16805 | positive | After viewing several episodes of this series,I have come to the conclusion that television producers··· |

removed RPS-$l_2$ from the candidate list as it requires fine-tuning the model with $l_2$ normalization, which is incompatible with tree ensemble models. Here, we highlight the following observations:

- The explanation produced by RPS-LJE is more similar to the test point than that of Influence Function in the sense of sharing similar feature values. For instance, the "Checking Account" values of the RPS-LJE explanation always align with the test samples, but that of the Influence Function does not.
- Influence Function tends to provide diverse explanations for each sample case, where the explanations produced by it show more or fewer differences.

## 5    Conclusion and Discussion

We presented an approach for explaining the impact of training data on a test prediction, called Representer Point Selection via Local Jacobian Expansion (RPS-LJE). Our approach aimed to correct two drawbacks of the existing Representer Point approach (RPS-$l_2$) [23], namely that it often 1) produces identical explanations for different instances in the same class and 2) produces highly varying explanations based on the strength of an $l_2$ regularization modification to the original model. We began by observing these problems empirically and then analyzing the RPS-$l_2$ derivation to reveal the technical source of these problems. We then proposed corrections to derive a novel form of RPS based on a local Jacobian Taylor expansion that addresses the technical limitations of the RPS-$l_2$.

We conducted multiple experiments that quantitatively and qualitatively analyzed the proposed RPS-LJE against existing state-of-the-art approaches, RPS-$l_2$ and Influence Function. Our experiments empirically show that the proposed RPS-LJE fulfilled our expectation in terms of correcting RPS-

Table 2: **Explanation Comparison between RPS-LJE and Influence Function on German Credit Data.** Column "Risk" is the target (label) column, and all columns after it are feature columns. Examples are randomly selected from the test set.

| | | ID | Risk | Checking Account | Credit History | Savings Account | Other Debtors | Employment |
|---|---|---|---|---|---|---|---|---|
| **Sample 1** | Test point | 318 | low | none | critical account/ other credits existing | little | none | 1 to 4 years |
| RPS-LJE | Explanation 1 | 210 | low | none | critical account/ other credits existing | unknown/none | none | 1 to 4 years |
| | Explanation 2 | 526 | low | none | critical account/ other credits existing | moderate | none | 1 to 4 years |
| | Explanation 3 | 294 | high | none | critical account/ other credits existing | unknown/none | none | more than 7 years |
| Influence Function | Explanation 1 | 668 | high | poor | critical account/ other credits existing | little | co-applicant | more than 7 years |
| | Explanation 2 | 747 | high | poor | existing credits paid back duly till now | little | none | less than 1 year |
| | Explanation 3 | 611 | high | moderate | existing credits paid back duly till now | moderate | none | more than 7 years |
| **Sample 2** | Test point | 414 | high | poor | existing credits paid back duly till now | unknown/none | none | 1 to 4 years |
| RPS-LJE | Explanation 1 | 828 | high | poor | existing credits paid back duly till now | unknown/none | none | more than 7 years |
| | Explanation 2 | 796 | high | poor | existing credits paid back duly till now | unknown/none | none | more than 7 years |
| | Explanation 3 | 918 | high | poor | existing credits paid back duly till now | moderate | none | unemployed |
| Influence Function | Explanation 1 | 828 | high | poor | existing credits paid back duly till now | unknown/none | none | more than 7 years |
| | Explanation 2 | 252 | high | little | existing credits paid back duly till now | little | guarantor | 1 to 4 years |
| | Explanation 3 | 796 | high | poor | existing credits paid back duly till now | unknown/none | none | more than 7 years |
| **Sample 3** | Test point | 951 | high | poor | delay in paying off in the past | little | none | 4 to 7 years |
| RPS-LJE | Explanation 1 | 174 | high | poor | delay in paying off in the past | little | none | less than 1 year |
| | Explanation 2 | 466 | high | poor | delay in paying off in the past | little | none | less than 1 year |
| | Explanation 3 | 862 | high | poor | existing credits paid back duly till now | little | none | less than 1 year |
| Influence Function | Explanation 1 | 174 | high | poor | delay in paying off in the past | little | none | less than 1 year |
| | Explanation 2 | 172 | high | little | delay in paying off in the past | little | none | unemployed |
| | Explanation 3 | 534 | low | none | existing credits paid back duly till now | unknown/ none | none | less than 1 year |

$l_2$'s problems. It produces individualized explanations instead of a class-level explanation and quantitatively performs comparably to, or outperforms, existing data-explanation approaches.

# 6 Scope of Application

This paper presents a novel method of sample-based model explanation for classifiers including deep neural networks and ensemble models. As stated in Section 3, RPS-LJE is applicable to classification models with a linear last layer before the activation function. While this requirement may seem restrictive, many widely-adopted classifiers satisfy them.

In Section 3.1, we state our assumption that the given model is well-trained (near the saddle point), and thus the gradient of loss with respect to the parameters is close to zero. RPS-$l_2$ [23] and Influence Function [11] also makes the same assumption. The assumption often holds in practice, as many classification models will be trained to near-convergence before deployment. However, one exception to this assumption would be when early stopping is used as a form of regularization.

# 7 Broader Impacts

Deep learning as well as ensemble methods such as XGBoost have shown exceptional performance in a variety of data-driven prediction applications including image classification, sentiment classification, and risk classification (cf. Section 4). This paper proposes a methodology to explain these complex classifiers via estimating the influence from each training datum on model predictions.

Sample-based explanation methods are critically important for validating and improving classifiers. For example, explanations can increase the interpretability and transparency of a model's decision making process, and thus help to assess the fairness of the model. Furthermore, monitoring training data quality can facilitate the debugging process and thus improve model performance. One potential negative impact is that the use of training data for explanations may raise privacy concerns in some situations; however, measures for data anonymization may (partially) help mitigate such issues.

## Acknowledgments and Disclosure of Funding

Yi Sui was funded by a University of Toronto Dean's Spark award to Scott Sanner. Ga Wu was funded by a Canadian NSERC Discovery Grant award to Scott Sanner.

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
