## A  Related Work

There are mainly three categories of methods for model explanation. The first one is feature-based that points out the important input features. LIME [16] explains the decision locally by approximating with an interpretable model. SHAP [13] uses the Shapley value as a measure of feature importance. The second category of methods is gradient-based, which are broadly used in explaining Convolutional Neural Networks (CNNs). These methods highlight the salient region of predictions with the input gradients [18, 20, 21, 3]. Some methods further analyze the effect of perturbing the network's input on its output [3, 5]. The last category of the methods is sample-based, where we attempt to understand the influence of the training data on test data predictions [1, 11, 17, 15, 8, 19].

## B  Detailed derivation of RPS-LJE

In this section, we provide a detailed derivation for Equation 11:

$$
\begin{aligned}
\Theta_L^\dagger \phi(\mathbf{x}_t) &= \Theta_L^* \phi(\mathbf{x}_t) - \sum_{i=1}^n \frac{1}{n} \mathcal{H}_{\Theta_L^*}^{-1} \nabla_{\Theta_L} \mathcal{L}(\mathbf{x}_i, y_i, \boldsymbol{\Theta}^*) \phi(\mathbf{x}_t) \\
&= \Theta_L^* \phi(\mathbf{x}_t) - \sum_{i=1}^n \left[ \frac{1}{n} \mathcal{H}_{\Theta_L^*}^{-1} \frac{\partial \mathcal{L}(\mathbf{x}_i, y_i, \boldsymbol{\Theta}^*)}{\partial \Theta_L^* \phi(\mathbf{x}_i)} \cdot \frac{\partial \Theta_L^* \phi(\mathbf{x}_i)}{\partial \Theta_L^*} \phi(x_t) \right] \\
&= \Theta_L^* \phi(\mathbf{x}_t) - \sum_{i=1}^n \left[ \frac{1}{n} \mathcal{H}_{\Theta_L^*}^{-1} \frac{\partial \mathcal{L}(\mathbf{x}_i, y_i, \boldsymbol{\Theta}^*)}{\partial \Theta_L^* \phi(\mathbf{x}_i)} \phi(x_i)^T \phi(x_t) \right] \\
&= \sum_{i=1}^n \frac{1}{n} \Theta_L^* \frac{1}{\phi(x_i)} \phi(x_i)^T \phi(\mathbf{x}_t) - \sum_{i=1}^n \left[ \frac{1}{n} \mathcal{H}_{\Theta_L^*}^{-1} \frac{\partial \mathcal{L}(\mathbf{x}_i, y_i, \boldsymbol{\Theta}^*)}{\partial \Theta_L^* \phi(\mathbf{x}_i)} \phi(x_i)^T \phi(x_t) \right] \\
&= \sum_{i=1}^n \underbrace{\left[ \Theta_L^* \frac{1}{\phi(\mathbf{x}_i)n} - \frac{1}{n} \mathcal{H}_{\Theta_L^*}^{-1} \frac{\partial \mathcal{L}(\mathbf{x}_i, y_i, \boldsymbol{\Theta}^*)}{\partial \Theta_L^* \phi(\mathbf{x}_i)} \right]}_{\boldsymbol{\alpha}_i} \underbrace{\phi(\mathbf{x}_i)^T \phi(\mathbf{x}_t)}_{\mathcal{K}(\mathbf{x}_i, \mathbf{x}_t)}
\end{aligned}
$$

The reciprocal of a vector is element-wise and $\partial \Theta_L^* \phi(\mathbf{x}_i)$ may be read as $\partial [\Theta_L^* \phi(\mathbf{x}_i)]$ for clarity.

## C  Explanation on Perturbed Samples

In this section, we further investigate the performance of Influence Function, RPS-$l_2$, and RPS-LJE by generating explanations for perturbed samples in the sentiment analysis task. Specifically, we perturb a training example by substituting a key word with its synonym (e.g. change "love" to "like") and observe the rank 1 explanations generated for the perturbed sample. As shown in Table 3, both Influence Function and RPS-LJE are able to rank the original training sample as the top explanation, whereas RPS-$l_2$ is not.

## D  Correlations between the Prediction Output and the Decomposition

In this section, we investigate the correlation between the prediction output $\hat{y}_t$ and the decomposition of the RPS methods $\sum_i \alpha_i K(x_t, x_i)$. As shown in Table 4, all correlation values are close to 1. Therefore, both RPS methods provide a highly correlated decomposition with respect to the actual prediction outputs.

## E  $l_2$ coefficient sensitivity of RPS-$l_2$

In this section, we explore the effect of the $l_2$ weights of RPS-$l_2$.

**Decomposition accuracy**
We calculate the Pearson correlations between the true prediction and the decomposition with RPS-$l_2$. As shown in Table 5, the decomposition accuracy is quite robust with respect to different $\lambda$ values. Therefore, in our experiments, we choose the $\lambda$ parameter within the common range of

Table 3: Explanations generated by Influence Function, RPS-$l_2$, and RPS-LJE for perturbed samples with LSTM on sentiment analysis task

| Data Type | ID | Raw Reviews |
|---|---|---|
| Original sample | 2619 | Wow, this was another good spin off of the original American pie··· |
| Perturbed sample | 2619 | Wow, this was another *great* spin off of the original American pie··· |
| Influence Function | 2619 | Wow, this was another good spin off of the original American pie··· |
| RPS-$l_2$ | 14701 | This is a very memorable spaghetti western.It has a great storyline··· |
| RPS-LJE | 2619 | Wow, this was another good spin off of the original American pie··· |
| Original sample | 4789 | Simply the best Estonian film that I have ever seen, although it is··· |
| Perturbed sample | 4789 | Simply the *greatest* Estonian film that I have ever seen, although it··· |
| Influence Function | 4789 | Simply the best Estonian film that I have ever seen, although it is··· |
| RPS-$l_2$ | 14701 | This is a very memorable spaghetti western.It has a great storyline··· |
| RPS-LJE | 4789 | Simply the best Estonian film that I have ever seen, although it is··· |
| Original sample | 11177 | I can't tell you all how much I love this movie. I have read reviews··· |
| Perturbed sample | 11177 | I can't tell you all how much I *like* this movie. I have read reviews··· |
| Influence Function | 11177 | I can't tell you all how much I love this movie. I have read reviews··· |
| RPS-$l_2$ | 9112 | Tim Krabbe is the praised author of 'Het Gouden Ei' , a novel that··· |
| RPS-LJE | 11177 | I can't tell you all how much I love this movie. I have read reviews··· |

Table 4: Pearson correlations between the prediction outputs and the decomposition generated by the RPS methods. Correlations are rounded down to two significant digits.

(a) Training samples

| Method | ResNet | LSTM |
|---|---|---|
| RPS-$l_2$ | 0.99 | 0.99 |
| RPS-LJE | 0.99 | 0.99 |

(b) Testing samples

| Method | ResNet | LSTM |
|---|---|---|
| RPS-$l_2$ | 0.99 | 0.99 |
| RPS-LJE | 0.99 | 0.98 |

the $l_2$ regularization coefficient from 1e-4 to 3e-3 (also the default $\lambda$ value from the RPS-$l_2$ public codebase).

Table 5: Pearson correlation between the actual prediction on the decomposition with different l2 coefficient $\lambda$ values with CIFAR-10 on ResNet-20 for RPS-$l_2$ (round down to 3 significant digits).

| $l_2$ coefficient value | $\lambda = 1e-5$ | $\lambda = 1e-4$ | $\lambda = 1e-3$ | $\lambda = 1e-2$ | $\lambda = 1e-1$ |
|---|---|---|---|---|---|
| Pearson correlation | 0.999 | 0.999 | 0.999 | 0.999 | 0.999 |

**Identical Explanation Issue (Shown in Figure 1 (b))**
We plotted the top-1 explanation generated by RPS-$l_2$ of four samples within the same class "cars". As shown in figure 6, all four samples share the same top-1 explanations in every $\lambda$ settings. Therefore, the identical explanation issue persists with different $\lambda$ values.

# F   Learning Rate Sensitivity of RPS-LJE

In this section, we further investigate the learning rate sensitivity of RPS-LJE.

From Figure 7 and Table 6, we can observe that RPS-LJE is quite robust with respect to different learning rate $\gamma$ values. Therefore, in our experiments, we picked learning rates within range of the common neural network training from $1e-5$ to $1e-2$.

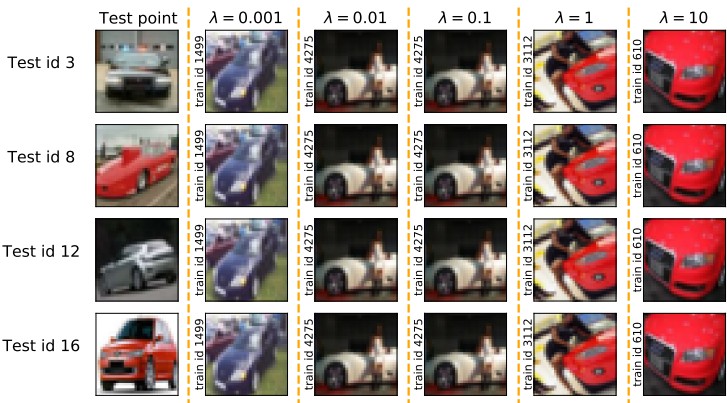

Figure 6: Top-1 explanation provided by RPS-$l_2$ with different $\lambda$ values.

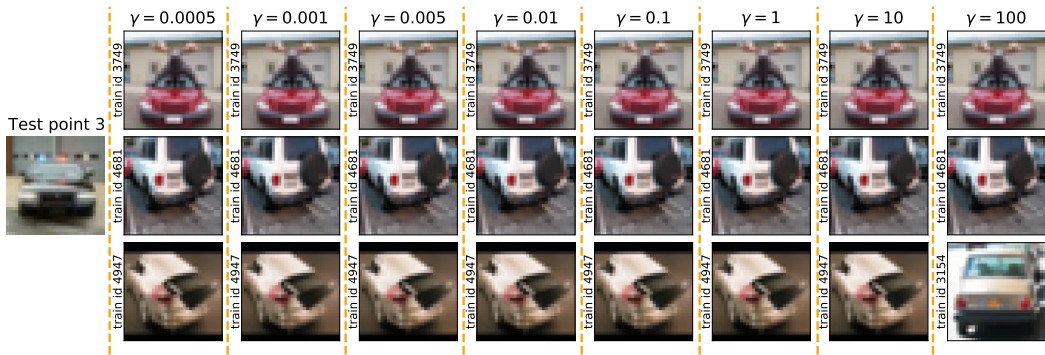

Figure 7: Explanation generated by RPS-LJE with different learning rate value for test sample with ID 3 on the IMDB dataset.

Table 6: Explanation generated by RPS-LJE with different learning rate $\gamma$ values for test sample with ID 5 on the IMDB dataset. Explanations are represented with their training sample ID.

| Learning rate value | $\gamma = 0.00001$ | $\gamma = 0.0001$ | $\gamma = 0.001$ | $\gamma = 0.01$ | $\gamma = 0.1$ | $\gamma = 1$ |
|---|---|---|---|---|---|---|
| Explanation 1 ID | 13580 | 13580 | 13580 | 13580 | 13580 | 7838 |
| Explanation 2 ID | 7838 | 7838 | 7838 | 7838 | 7838 | 13580 |
| Explanation 3 ID | 7322 | 7322 | 7322 | 7322 | 7322 | 7322 |

# G Correlation with Influence Function

In this section, we further explore the similarity and difference between RPS-LJE and Influence Function. Therefore, we compute Pearson and Spearman correlations of the top 5 explanation between RPS-LJE and Influence Functions (we take the union of the top-5 explanations from the two methods). The Pearson correlations are shown in Table 7 and the Spearman correlations are show in Table 8.

Our results show, while in many cases the two approaches provide similar explanations, there are cases where explanations are completely different. This observation reflects the fundamental difference between equation (11) and (13), where equation (11) includes an additional factor and equation (13) includes a multiplicative factor. This experiment demonstrates that even though the two approaches appear similar, the subtle difference still makes the two approaches show different behaviour, even in practice.

Table 7: Pearson correlations between the top-5 explanations from RPS-LJE and Influence Functions. (We report the distributions in quantiles)

| Quantile of distribution | 0.01% | 0.1% | 1% | 10% | 30% | 50% | 70% | 90% |
|---|---|---|---|---|---|---|---|---|
| ResNet-20 | 0.9998 | 0.9998 | 0.9999 | 0.9999 | 0.9999 | 0.9999 | 0.9999 | 0.9999 |
| Bi-LSTM | 0.1484 | 0.7788 | 0.9513 | 0.8855 | 0.9992 | 0.9997 | 0.9999 | 0.9999 |
| XGBoost | -0.8628 | -0.8628 | -0.7201 | -0.5694 | -0.2998 | -0.0569 | 0.2451 | 0.6025 |

Table 8: Spearman correlations between the top-5 explanations from RPS-LJE and Influence Functions. (We report the distributions in quantiles)

| Quantile of distribution | 0.01% | 0.1% | 1% | 10% | 30% | 50% | 70% | 90% |
|---|---|---|---|---|---|---|---|---|
| ResNet-20 | 0.7000 | 0.9000 | 0.9428 | 1.0000 | 1.0000 | 1.0000 | 1.0000 | 1.0000 |
| Bi-LSTM | 0.1000 | 0.4000 | 0.7000 | 0.9000 | 1.0000 | 1.0000 | 1.0000 | 1.0000 |
| XGBoost | -0.8214 | -0.8214 | -0.8181 | -0.6666 | -0.3833 | -0.1428 | 0.0714 | 0.4285 |

## H  Detailed Experimental setup

This section provides information on the detailed experimental setup.

### H.1  Training details

Similar to [15], we calculate Influence Function with respect to the parameters in the last linear layer (and consider the previous layers frozen) as an approximation, since the computation with respect to all parameters is prohibitively expensive.

**CIFAR-10 Image Classification with ResNet**
For all CIFAR models(car vs horse), we use a 5:1 train-test split and fine-tune on a pre-trained ResNet (with 92.6% test accuracy on CIFAR-10) with 272K parameters.

In the data debugging experiment(Section 4.2), the testing accuracy drops from 100% to 89% after data corruption. While computing the self-prediction contribution, we use $1e-3$ as the $l_2$ coefficient for RPS-$l_2$. For RPS-LJE, we use $1e-5$ as the learning rate of the one-step gradient ascent. For every 5% of data checked, we retrain the ResNet three times using a SGD optimizer with learning rate 0.01 for 30 epochs and report the average test accuracy.

For understanding model behaviour, we fine-tune the ResNet using a SGD optimizer with learning rate of 0.01 for 20 epochs. The test accuracy is 100%. To generate explanations, for RPS-$l_2$, we use $1e-4$ as the $l_2$ coefficient. For RPS-LJE, we use $1e-2$ as the learning rate of the one-step gradient ascent.

**IMDB Sentiment Analysis with LSTM**
For all IMDB models, we use a 7:3 train-validation split and 1:1 train-test split to train a 2-layer Bi-LSTM(4.81M parameters) with Glove embedding. The vocabulary size is 25K.

In the data debugging experiment, the testing accuracy drops from 89.8% to 70.9% after data corruption. While calculating the self-prediction contribution, for RPS-$l_2$, we use $3e-3$ as the $l_2$ coefficient. For RPS-LJE, we use $1e-5$ as the learning rate of the one-step gradient ascent. For every 5% of data checked, we retrain the LSTM two times using an Adam optimizer with learning rate of $5e-3$ for 20 epochs and report the average test accuracy.

For understanding model behaviour, we train the LSTM with an Adam optimizer with learning rate of $1e-3$ and train for 10 epochs. The test accuracy of the trained LSTM is 89.8%. To generate explanations, for RPS-$l_2$, we use $3e-3$ as the $l_2$ coefficient. For RPS-LJE, we use $1e-5$ as the learning rate of the one-step gradient ascent.

**German Credit Risk Analysis with XGBoost**
For all German Credit Risk models, We use a 4:1 train-test split. In the data debugging experiment, the testing accuracy drops from 78.5% to 73.4% after data corruption. While calculating the self-

prediction contribution we use $1e - 5$ as the learning rate of the one-step gradient ascent for RPS-LJE. For every 5% of data checked, we retrain the XGBoost model five times and report the average test accuracy.

For understanding model behaviour, we use $1e - 4$ as the learning rate of the one-step gradient ascent for RPS-LJE.

## H.2 Understanding model behaviour

In the experiment of understanding model behaviour (Section 4.3), we compared explanations for test points from Influence Function, RPS-$l_2$, and RPS-LJE across different tasks. The training details of models are provided in the previous section. Since the training accuracies of the IMDB sentiment analysis model and the German Credit risk analysis model are both below 100%, we adopt the same strategy as in [15] to generate meaningful explanations. Specifically, for all explanation methods, we excluded the wrongly predicted training data points, considering that they naturally have large gradients.

## H.3 Experiment Platform

In our work, we conduct the experiments on two workstations: one for sentiment classification task with Bi-LSTM, and the other for Image classification with ResNet and German Credit analysis with XGBoost. The workstations configurations is shown in Table 9. The softwares used for conducting the experiments are listed in table 10.

Table 9: Summary of computational resource

| Experiment models | Memory | Hard drive | CPU | GPU |
|---|---|---|---|---|
| Bi-LSTM | 64GB | 512GB SSD+2TB HDD | Intel Core i7-10700 | Nvidia RTX3090 |
| ResNet and XGBoost | 64GB | 1TB SSD+4TB HDD | Intel Corei7-9700K | GeForce RTX3080 |

Table 10: Software versions

| Experiment models | Python | Pytorch | Xgboost |
|---|---|---|---|
| Bi-LSTM | 3.8.8 | 1.7.1 | - |
| ResNet and XGBoost | 3.6.9 | 1.7.0 | 1.4.2 |

# I   Additional experiment result

This section provides additional experiment results to support the main body of the paper. Note, the observations here align with the observations/conclusions in the main paper.

**German Credit Risk Analysis with XGBoost**

Table 11 lists some additional result on German Credit Risk model(XGBoost). With these samples, we make the following observations:

- RPS-LJE and Influence Function agree on some explanations (1 or 2 out of 3).
- RPS-LJE tends to provide more coherent explanations to the original test sample compared to Influence Function. For example, the features of RPS-LJE's explanations for test Sample 1 all have "Savings Account" as little, where as Influence Function's explanation is more diverse. Also, RPS-LJE's Explanations all have the same risk level with the original test sample, whereas Influence Function's generates explanations with different risk level from the original test sample for Sample 2 (Explanation 2) and Sample 3 (Explanation 1 and 3).

**CIFAR-10 Image Classification with ResNet**

Table 11: **Explanation Comparison between RPS-LJE and Influence Function on German Credit Data.** Column "Risk" is the target (label) column, and all columns after it are feature columns. Examples are randomly selected from the test set.

| | | ID | Risk | Checking Account | Credit History | Savings Account | Other Debtors | Employment |
|---|---|---|---|---|---|---|---|---|
| **Sample 1** | Test point | 885 | high | poor | existing credits paid back duly till now | little | none | less than 1 year |
| RPS-LJE | Explanation 1 | 471 | high | poor | existing credits paid back duly till now | little | none | less than 1 year |
| | Explanation 2 | 862 | high | poor | existing credits paid back duly till now | little | none | less than 1 year |
| | Explanation 3 | 649 | high | poor | existing credits paid back duly till now | little | none | 1 to 4 years |
| Influence Function | Explanation 1 | 471 | high | poor | existing credits paid back duly till now | little | none | less than 1 year |
| | Explanation 2 | 610 | high | poor | existing credits paid back duly till now | moderate | none | unemployed |
| | Explanation 3 | 203 | high | poor | existing credits paid back duly till now | little | none | 4 to 7 years |
| **Sample 2** | Test point | 507 | high | little | all credits at this bank paid back duly | moderate | none | unemployed |
| RPS-LJE | Explanation 1 | 594 | high | poor | all credits at this bank paid back duly | unknown/none | none | more than 7 years |
| | Explanation 2 | 583 | high | little | existing credits paid back duly till now | little | none | less than 1 year |
| | Explanation 3 | 182 | high | poor | all credits at this bank paid back duly | unknown/none | none | 1 to 4 years |
| Influence Function | Explanation 1 | 583 | high | little | existing credits paid back duly till now | little | none | less than 1 year |
| | Explanation 2 | 712 | low | none | existing credits paid back duly till now | unknown/none | none | more than 7 years |
| | Explanation 3 | 594 | high | poor | all credits at this bank paid back duly | unknown/none | none | more than 7 years |
| **Sample 3** | Test point | 744 | low | poor | critical account/ other credits existing | unknown/none | none | 4 to 7 years |
| RPS-LJE | Explanation 1 | 654 | low | none | critical account/ other credits existing | little | none | 4 to 7 years |
| | Explanation 2 | 380 | low | poor | existing credits paid back duly till now | unknown/none | none | 4 to 7 years |
| | Explanation 3 | 712 | low | none | existing credits paid back duly till now | unknown/none | none | more than 7 years |
| Influence Function | Explanation 1 | 471 | high | poor | existing credits paid back duly till now | little | none | less than 1 year |
| | Explanation 2 | 380 | low | poor | existing credits paid back duly till now | unknown/none | none | 4 to 7 years |
| | Explanation 3 | 289 | high | poor | delay in paying off in the past | little | none | less than 1 year |

Figure 8: **Comparison of Top-4 Influential Training Images for Three Test Samples.** For each test sample, the upper row shows positive influential points, whereas the bottom row shows negative influential points. Examples are randomly selected from 2000 test samples in our experiment setting.

Figure 8 shows some additional experiment results for CIFAR-10 Image Classification. In these examples, we further confirm our observations from Figure 5:

- Influence Function and RPS-LJE provide similar explanations, which reflects the similarity we identified in their formula of computing training data importance in Section 3.3.
- RPS-$l_2$ repeatedly provides similar explanations to the test points in the same class, which aligns with our observations in Figure 5 and Figure 1. Particularly, the 3 test points in Figure 8 share 3 out of 4 positive explanations(explanation 1, 2, and 4; training example ID-6793, ID-7655, and ID-5424).

**IMDB Sentiment Analysis with LSTM**

Table 12 displays additional results on the IMDB sentiment analysis model (Bi-LSTM). Here, we confirm our previous observations:

- RPS-LJE and Influence Functions generally agree on the explanations(2 out of top 3). On the different explanation training point.
- RPS-LJE provides more coherent explanations in terms of narrative styles. For instance, all explanations provided by RPS-LJE for Sample 1 starts with "If you".

Table 12: **Explanation Comparison among RPS-LJE, RPS-$l_2$ and Influence Function on IMDB sentiment analysis data.** Column "Sentiment" is the target (label) column, and raw review is the input of Bi-LSTM network. Examples are randomly selected from the test set.

| | | ID | Sentiment | Raw Reviews |
|---|---|---|---|---|
| **Sample 1** | Test point | 293 | positive | If you need that instant buzz that only late 60s/early 70s Euro sex movies can give off, then look⋯ |
| RPS-LJE | Explanation 1 | 1091 | positive | If you "get it", it's magnificent. If you don't, it's decent. Please understand that "getting it" does⋯ |
| | Explanation 2 | 7896 | positive | If you have any clue about Jane Austen´s production, you´ll now that she repeats the same in each⋯ |
| | Explanation 3 | 1216 | positive | If you fast forward through the horrible singing,you will find a classic fairy tale underneath.Chris⋯ |
| RPS-$l_2$ | Explanation 1 | 9112 | positive | Tim Krabbe is the praised author of 'Het Gouden Ei' , a novel that was put on the screen twice⋯ |
| | Explanation 2 | 3704 | positive | THE DEVIL'S PLAYTHING is my second attempt at a Joseph Sarno production-and although I ⋯ |
| | Explanation 3 | 4000 | positive | So , Todd Sheets once stated that he considers his 1993 , shot-on-video Z - epic, Zombie Bloodbath⋯ |
| Influence function | Explanation 1 | 1091 | positive | If you "get it", it's magnificent. If you don't, it's decent. Please understand that "getting it" does⋯ |
| | Explanation 2 | 7896 | positive | If you have any clue about Jane Austen´s production, you´ll now that she repeats the same in each⋯ |
| | Explanation 3 | 13487 | positive | It surprises me that I actually got the courage to watch the bio flick or flicks "Che : Parts 1 & 2"⋯ |
| **Sample 2** | Test point | 450 | positive | I can't praise this film enough.It had a lot of that hand-held, first-person shaking camera which I⋯ |
| RPS-LJE | Explanation 1 | 3342 | positive | I barely remember this show,a little,but I remembered it was great!My eldest brother, reminded me⋯ |
| | Explanation 2 | 3351 | positive | I simply love this movie. It is a perfect example of the well-rounded surprising stories that come⋯ |
| | Explanation 3 | 11256 | positive | I watched this flick yesterday and I have to say it's the finest horror film made for$36,000 I've ever⋯ |
| RPS-$l_2$ | Explanation 1 | 9112 | positive | Tim Krabbe is the praised author of 'Het Gouden Ei' , a novel that was put on the screen twice⋯ |
| | Explanation 2 | 3704 | positive | THE DEVIL'S PLAYTHING is my second attempt at a Joseph Sarno production-and although I ⋯ |
| | Explanation 3 | 4000 | positive | So , Todd Sheets once stated that he considers his 1993 , shot-on-video Z - epic, Zombie Bloodbath⋯ |
| Influence function | Explanation 1 | 3342 | positive | I barely remember this show,a little,but I remembered it was great!My eldest brother, reminded me⋯ |
| | Explanation 2 | 11256 | positive | I watched this flick yesterday and I have to say it's the finest horror film made for$36,000 I've ever⋯ |
| | Explanation 3 | 3957 | positive | What an amazing film. With very little dialogue, the whole story is told with glances and body⋯ |
| **Sample 3** | Test point | 13147 | negative | The movie starts with a Spiderman spoof which is your introduction to Rick Riker(played by Drake⋯ |
| RPS-LJE | Explanation 1 | 172 | negative | The day has finally come for me to witness the perpetuation of Azumi's fate as an assassin, fruition⋯ |
| | Explanation 2 | 16133 | negative | The Wicker Man, starring Nicolas Cage, is by no means a good movie, but I can't really say it's⋯ |
| | Explanation 3 | 11928 | negative | The Lives of the Saints starts off with an atmospheric vision of London as a bustling city of busy⋯ |
| RPS-$l_2$ | Explanation 1 | 4801 | negative | A so common horror story about a luxury building at Brooklyn which hides the gates to hell . It is⋯ |
| | Explanation 2 | 11015 | negative | The thing that stands out in my mind in this film ( sadly ) is the introduction , where John Berlin⋯ |
| | Explanation 3 | 12446 | negative | Taped this late night movie when I was in grade 11 , watched it on fast forward . I sugest you do⋯ |
| Influence function | Explanation 1 | 172 | negative | The day has finally come for me to witness the perpetuation of Azumi's fate as an assassin, fruition⋯ |
| | Explanation 2 | 16133 | negative | The Wicker Man, starring Nicolas Cage, is by no means a good movie, but I can't really say it's⋯ |
| | Explanation 3 | 8208 | negative | For all its visual delights, how much better Renaissance would have been in live action.The anim-⋯ |

- RPS-$l_2$ provides exactly the same explanations for the two samples with positive sentiment(Sample 1 and Sample 2). This observation aligns with our findings in Figure 1, where RPS-$l_2$ appears to be more of a class-level explanation method rather than instance-level explanation method.