# OpenReview forum: "Representer Point Selection via Local Jacobian Expansion for Post-hoc Classifier Explanation of Deep Neural Networks and Ensemble Models"
_NeurIPS.cc/2021/Conference — NeurIPS 2021 Poster_

### Official Review · Reviewer_3a5c · 2021-07-12

**Rating:** 7
**Confidence:** 3

**Summary:**

The aim of this paper is to identify the contribution of the training data to the prediction. This work highlights the short comings of the existing Represented Point Selection (RPS-$l_2$) both empirically and by analyzing the derivation of RPS$-l_2$, and proposes an improved method called RPS-LJE. RPS-LJE is a clear improvement over RPS-$-l_2$ both qualitatively and quantitatively. Qualitatively, it is observed that the explanations produced by RPS-LJE isn't repetitive like RPS$-l_2$. Quantitatively, it performs comparably to, or better that the existing data-explanation approaches.

**Ethical Concerns:**

Nil.

**Limitations And Societal Impact:**

Nil.

**Main Review:**

Originality: The approach presented in this work to address the shortcomings of RPS-$l_2$ is definitely novel

Please consider citing the following saliency methods for model explanation  -
https://arxiv.org/pdf/2006.16322.pdf
https://arxiv.org/abs/1703.01365

Quality:
The work is technically sound. The derivation of the RPS-LJE is pretty neat and intuitive. The effectiveness of the algorithm is further demonstrated empirically by the experiments where we see both qualitative and quantitative improvements.

Clarity: This work is well written and well organized.

Significance:
The results are significant.

----------------------
Post Rebuttal Comments:
After going through the rebuttal I still believe that the paper is an accept, though not a *clear accept*, and hence, I'm updating my score to **7**.

Some comments for a future version of the paper:

I agree with DC2m that the (mean) test scores as reported aren't meaningful enough to make a quantitative argument that RPS-LJE is better than the baselines. However, the authors' response seems valid as well, i.e., the std is primarily influenced by the randomness of data corruptions across multiple runs. In such a case, the authors could report the fraction of examples for which RPS-LJE achieves the best score. Comparing the mean and std of this metric would be more meaningful.

Also, do consider citing: [Scaling Symbolic Methods using Gradients for Neural Model Explanation](https://arxiv.org/pdf/2006.16322.pdf)

**Time Spent Reviewing:**

1 hour

---

> ### Author Response · Authors · 2021-08-10
> **response to reviewer 3a5c**
>
> Thanks for your feedback and for pointing out the related work. We will add in the reference of the suggested paper as one of the saliency methods in line 26.  Moreover, we will add the following related work section thanks to the suggestion by reviewer Dc2m:
>
> There are mainly three categories of methods for model explanation. The first one is feature-based that points out the important input features. LIME[12] explains the decision locally by approximating with an interpretable model. SHAP[9] uses the Shapley value as a measure of feature importance. The second category of methods is gradient-based, which are broadly used in explaining Convolutional Neural Networks (CNNs). These methods highlight the salient region of predictions with the input gradients [14, 15, 18, 19]. Some methods further analyze the effect of perturbing the network’s input on its output [19, 20]. The last category of the methods is sample-based, where we attempt to understand the influence of the training data on test data predictions [1, 7, 11, 17, 22, 23].
>
> [18] Sundararajan, Mukund & Taly, Ankur & Yan, Qiqi. (2017). Axiomatic Attribution for Deep Networks.
>
> [19] Dabkowski, Piotr & Gal, Yarin. (2017). Real Time Image Saliency for Black Box Classifiers.
>
> [20] Fong, Ruth & Vedaldi, Andrea. (2017). Interpretable Explanations of Black Boxes by Meaningful Perturbation.
>
> [21] Fong, Ruth & Patrick, Mandela & Vedaldi, Andrea. (2019). Understanding Deep Networks via Extremal Perturbations and Smooth Masks.
>
> [22] Khanna, Rajiv & Kim, Been & Ghosh, Joydeep & Koyejo, Oluwasanmi. (2018). Interpreting Black Box Predictions using Fisher Kernels.
>
> [23] Sharchilev, B., Yury Ustinovsky, P. Serdyukov and M. Rijke. (2018). Finding Influential Training Samples for Gradient Boosted Decision Trees.
>
> ([18] is the suggested paper)
>
> We really hope our response addresses your concern. If you have any further questions, we would be happy to answer them in follow-up discussion.

---

### Official Review · Reviewer_DC2m · 2021-07-16

**Rating:** 5
**Confidence:** 3

**Summary:**

This paper addresses the problem of model explainability by looking at which training examples were responsible for the model’s predictions. The contributed model takes inspiration from RPS and changes its computation of the data importance factor such that it becomes less sensitive to the choice of hyper-parameters and leads to higher diversity in the explanations. The proposed RPS-LJE is compared against two baseline approaches from the state of the art.

**Limitations And Societal Impact:**

The paper discusses the broader impact of their work in the appendix. Although the checklist indicates that the limitations are discussed in the appendix, the reviewer could not find this discussion. The reviewer would strongly encourage the authors to move the broader impact to the main body of the paper. The reviewers would strongly encourage the authors to include a limitations discussion, also in the main body of the paper. This is crucial information that should not be disregarded.

What are the limitations of the proposed approach? It would seem that one of the limitations is that it would require storing all training data. Could the authors comment on that? Could we consider storing a subset of the training data that would be sufficient to provide explanations? How could we select the data points to consider?

**Main Review:**

**Originality:**

The proposed approach seems of moderate originality. It builds on top of RPS and changes the way it computes data importance factors. In particular, the proposed computation of the data importance term is rewritten as a gradient step with a dynamic learning rate. The resulting formulation becomes, as acknowledged in the paper, similar to the influence function method. Moreover, there is no related work section. While this could be ok, the related work covered in the submission appears to be rather limited. Positioning the contribution wrt a larger body of literature would strengthen the contribution. A few works that the authors might want to consider:
- https://arxiv.org/pdf/1810.10118.pdf
- https://arxiv.org/abs/1802.06640

and with a different flavor:
- https://arxiv.org/abs/1412.6806
- https://arxiv.org/abs/1704.03296
- https://arxiv.org/abs/1910.08485
- https://proceedings.neurips.cc/paper/2017/file/0060ef47b12160b9198302ebdb144dcf-Paper.pdf

**Quality and significance:**

The submission appears technically sound. The proposed approach is tested on three data debugging tasks (1 model per dataset) and compared against 2 baselines from the state of the art (RPS and influence function) and a random baseline. The reviewer's main concerns are wrt the experimental validation of the proposed approach and its motivation. See below.

- The choice of debugging tasks is unclear. Why train on a binary CIFAR-10 version (cars vs horses) as opposed to the standard `multi-class setup?
- The choice of methods to compare against is unclear and seems rather limited. Why limit the comparison to RPS and influence function? Why not consider [11] as well? Why is RPS removed from the German Credit Risk Analysis task? Why not use a model which is compatible with the RPS baseline?
- It appears that the experimental setup used in the paper is different than the one in the RPS paper. The paper considers corrupting 20-30% of the data, whereas 40% is corrupted in RPS. Why not stick to 40%?
- The paper claims that the proposed approach leads to more diverse explanations than RPS and shows this qualitatively. Would it be possible to quantify the diversity achieved by the proposed approach? e.g. by computing the % of explanations that are different from each other among the retrieved ones.
- The paper claims that the proposed approach leads to explanations that are faithful with the original model; however, the statement is only vaguely backed up. Could the authors quantify the agreement between original model output and the updated one?  (e.g. correlation) and compare the agreement with the one of other methods?
- Results are averaged across different runs but only the mean appears on the plots. How do the std look like? Are the observed improvements significant?
- What is the effect of considering learning rates even bigger than 1e-2? or in other words, how big does the learning rate have to be to start observing differences in the selected explanations?
- If we consider two images from the same class with similar appearances, do they lead to the same explanations?
- It seems that a natural baseline to the data debugging task would be to use the entropy at the output of a model to decide which data points to correct.
- Some experimental details are missing e.g.: which ResNet version is used? How is the RPS lambda chosen when making comparisons? How is the learning rate required by the proposed approach chosen?




- The paper does not motivate why explainability approaches which rely on understanding the influence of the training data on the test predictions are important.
- The paper does not motivate why explaining the predictions of a model is important.
- It is not clear why diversity in the explanations would be a desirable feature to improve the trustworthiness of a system. Could the authors elaborate on this?
- Could the authors comment on the use cases of the proposed approach? If the system fails, what happens? How can we mitigate this?
- What conclusions can we draw about the trustworthiness of the models using the proposed approach? What do we learn from those models that we couldn't learn from previous art?

**Clarity:**

Overall the paper is well written and easy to follow. Please find below some minor suggestions to improve its presentation.
- Figure 1: Include values of l2 weight. Does the observation about diversity hold for different l2 weights?
- Figure 2: Include details such as model used.
- Figure 3(b): Include choices of learning rates.
- Figure 4 contains a lot of plots. When discussing the figure, it would be good to point the reader towards each specific subplot from which conclusions are being drawn.
- Figure 5: It could be interesting to share associated importance values.
- The model notation introduced in Section 3 seems unnecessary.
- Introduce n (line 58).
- What is CNN in line 189 (the ResNet model?), same with RNN (the previously introduced LSTM?), it would be better if the authors could stick to the introduced names
- Although intuitive, random baseline should be explained.
- Appendix is rather packed, it would be helpful to refer to different relevant sections of the appendix in the main body of the paper.

**Comments about the checklist:**
- The code is not shared. Please revisit 3(a) and 4(c) and indicate that the code *will be* shared.
- Please review 4(b) and 4(d). Even if datasets are publicly available, they might come with their licenses, and unfortunately it is not always the case that consent was gathered from people.

**Post-rebuttal comments:**

The reviewer would like to thank the authors for their rebuttal which addressed a number of the questions raised in the reviews. After careful consideration and discussion with other reviewers, the reviewer agrees that the goal of the paper is sounds and that the proposed approach has some edge of novelty. Bridging IF and RPS-LJE may be of interest, although as acknowledged in the rebuttal, the benefits of the diversity achieved by the proposed approach remain unclear.


The remaining main concern lies in the execution, and in particular the experimental evidence to support the claims and to ensure robustness of the conclusions of the paper. Unfortunately, the rebuttal did not take the opportunity to discuss the statistical significance of the results. The reviewer agrees that the variance of all methods might come from the randomness of data corruptions, but still thinks it would be worth understanding how robust each approach might be wrt this randomness. Moreover, the authors chose a different experimental setup than the one in previous art. While this might be ok, the decision still seems arbitrary. As suggested in the rebuttal, increasing the % of corruption might make differences between the random baseline and other approaches smaller, but considering different % could perhaps be useful to paint the full picture and potentially lead to more insightful discussion of the results. Beyond these points, unfortunately the rebuttal did not take the opportunity to discuss potential use cases/failure modes of the proposed approach, very briefly surfaced some of the limitations pointed by the reviewers, and did not comment on the conclusions that can be drawn from this research.

Those are the main reasons why the reviewer remains on the fence and decides to keep their original score.


**Time Spent Reviewing:**

3h

---

> ### Author Response · Authors · 2021-08-10
> **response to reviewer DC2m [1/2]**
>
> Thanks for your feedback. We are happy to respond to your comments and questions as below:
>
> ___Related works___
>
> **The related work covered in the submission appears to be rather limited.**
>
> Thanks for your suggestion. Due to paper length limitations, we tried to merge the most significant related work into the introduction. However, we do plan to add the following related work section (perhaps in the appendix if needed):
>
> There are mainly three categories of methods for model explanation. The first one is feature-based that points out the important input features. LIME[12] explains the decision locally by approximating with an interpretable model. SHAP[9] uses the Shapley value as a measure of feature importance. The second category of methods is gradient-based, which are broadly used in explaining Convolutional Neural Networks (CNNs). These methods highlight the salient region of predictions with the input gradients [14, 15, 18, 19]. Some methods further analyze the effect of perturbing the network’s input on its output [19, 20]. The last category of the methods is sample-based, where we attempt to understand the influence of the training data on test data predictions [1, 7, 11, 17, 22, 23].
>
> [18] Sundararajan, Mukund & Taly, Ankur & Yan, Qiqi. (2017). Axiomatic Attribution for Deep Networks.
>
> [19] Dabkowski, Piotr & Gal, Yarin. (2017). Real Time Image Saliency for Black Box Classifiers.
>
> [20] Fong, Ruth & Vedaldi, Andrea. (2017). Interpretable Explanations of Black Boxes by Meaningful Perturbation.
>
> [21] Fong, Ruth & Patrick, Mandela & Vedaldi, Andrea. (2019). Understanding Deep Networks via Extremal Perturbations and Smooth Masks.
>
> [22] Khanna, Rajiv & Kim, Been & Ghosh, Joydeep & Koyejo, Oluwasanmi. (2018). Interpreting Black Box Predictions using Fisher Kernels.
>
> [23] Sharchilev, B., Yury Ustinovsky, P. Serdyukov and M. Rijke. (2018). Finding Influential Training Samples for Gradient Boosted Decision Trees.
>
> ___Quality and significance___
>
> **The choice of debugging tasks is unclear. Why train on a binary CIFAR-10 version (cars vs horses) as opposed to the standard `multi-class setup?**
>
> In the mislabelling debugging experiment, we followed the setup of prior works (RPS-$l_2$ [17] and Influence Function [7]) for consistency, where binary classification was used to demonstrate the debugging ability of candidate models. In particular, using automobiles and horses was suggested in RPS [17] (section 4.1).
>
> **The choice of methods to compare against is unclear and seems rather limited. Why limit the comparison to RPS and influence function? Why not consider [11] as well? Why is RPS removed from the German Credit Risk Analysis task? Why not use a model which is compatible with the RPS baseline?**
>
> Thanks for the comment. As stated in section 4(line 163), the method in [11] requires access to the full training history of the given model, which is out of the scope of the comparison in this paper. Specifically, we focus on post-training prediction explanations, where access to model training is not assumed (which can often be the case in practice where the downstream user of the model may not have access to the full training process).
>
> We cannot include RPS-$l_2$ for German Credit Risk Analysis task since the extra fine-tuning step with $l_2$ regularization is not compatible with the Xgboost model. We compare the proposed RPS-LJE to RPS-$l_2$ with both ResNet and LSTM. The purpose of the Xgboost experiment is to demonstrate the proposed RPS-LJE can also explain Xgboost classification models.
>
> **It appears that the experimental setup used in the paper is different than the one in the RPS paper. The paper considers corrupting 20-30% of the data, whereas 40% is corrupted in RPS. Why not stick to 40%?**
>
> Thanks for the comment. There is no universal setting for the data debugging experiment for now. The original Influence Function paper [7]  flips 10% of the data. The RPS-$l_2$ paper flips 40% of the data. Here, we choose the middle range of 20-30%. Also, we noticed in our experiment that when flipping 40% of the data, the performance between random baseline and all methods(Influence Function, RPS-$l_2$, RPS-LJE) is not obvious, which agrees with the trend shown in the RPS-$l_2$ paper. When the data has 40% of false labels, the classifiers might be too confused to make meaningful predictions.
>
> **The paper claims that the proposed approach leads to more diverse explanations than RPS and shows this qualitatively. Would it be possible to quantify the diversity achieved by the proposed approach? e.g. by computing the % of explanations that are different from each other among the retrieved ones.**
>
> Thanks for the suggestion. As shown in section 4, our proposed RPS-LJE method demonstrates a diverse collection of explanations compared to the RPS-$l_2$ method. However, we acknowledge that large diversity __alone__ does not guarantee good explanations and we do not optimize on diversity. Instead, we intend to provide instance-level explanations as opposed to the class-level explanation by RPS-$l_2$. Additionally, there is no well established metric for measuring the diversity.
>
> **The paper claims that the proposed approach leads to explanations that are faithful with the original model; however, the statement is only vaguely backed up. Could the authors quantify the agreement between original model output and the updated one? (e.g. correlation) and compare the agreement with the one of other methods?**
>
> Thanks for the comment. Unlike RPS-$l_2$, our proposed method does not require a fine-tuning step and thus does not alter the original model. As shown in equation 11, we only need a nearby anchor point $\Theta^\*$ in our result for the importance calculation. Theoretically, as long as $\Theta^\*$ is close to $\Theta^\dagger$, the Taylor expansion in equation 8 holds. Empirically, we have also demonstrated in Figure 2(b) that our method is quite robust with respect to the a large range of learning rate(up to $1e-2$) we choose to find $\Theta^*$.
>
> To further check the agreement, we conducted the following experiment of Pearson correlation between the decomposition and the prediction:
>
> Here is the result for training data(round down to keep 2 significant digits):
>
> | Method  	| ResNet | LSTM  |
> |-----------|--------|-------|
> | RPS-$l_2$ | 0.99   | 0. 99 |
> | RPS-LJE 	| 0.99   | 0. 99 |
>
> Here is the result for testing data(round down to keep 2 significant digits):
>
> | Method    | ResNet | LSTM |
> |-----------|--------|------|
> | RPS-$l_2$ | 0.99   | 0.99 |
> | RPS-LJE   | 0.99   | 0.98 |
>
> All correlation values are greater than 0.98.
>
> **Results are averaged across different runs but only the mean appears on the plots. How do the std look like? Are the observed improvements significant?**
>
> Thanks for this comment. We plotted the average following the past papers(Influence Function and RPS-$l_2$). Indeed, we noticed large variance (for all candidate explanation models) in our experiment. However, the variance comes from the randomness of data corruptions across multiple runs. It is not the variance of the explanation models. In particular, when the data corruption is fixed, all of the three models show stable performance with negligible variance to show.
>
> **What is the effect of considering learning rates even bigger than 1e-2? or in other words, how big does the learning rate have to be to start observing differences in the selected explanations?**
>
> Thanks for asking these questions. In our original experiment, we tune the learning rates in a range up to $1e-2$ because the Taylor expansion in our derivation requires $\Theta^\dagger$ to be sufficiently close to $\Theta^*$ (equation 8). Theoretically, an even larger learning rate may negatively impact the performance. Actually, we note that $[1e-5,...,1e-2]$ is a common range for training neural networks, and our proposed method is robust to this large range.
>
> To answer your second question, we have conducted an additional experiment to test on large learning rates with ResNet. Results are shown below. For this specific model, our method provides consistent explanations even with learning rate as 10 and changes when the learning rate is 100.
>
> Top 3 explanations of test sample with ID 3 (the numbers in the table are the IDs of the explanation training samples):
>
> |        | $lr=0.1$ | $lr=1$   | $lr=10$ | $lr=100$ |
> |--------|----------|----------|---------|----------|
> | Rank 1 | 3749     | 3749     | 3749    | 3749     |
> | Rank 2 | 4681     | 4681     | 4681    | 4681     |
> | Rank 3 | 4947     | 4947     | 4947    | 3154     |
>
> We further experiment with the LSTM model. The explanations are consistent up to 0.1 and change when the learning rate is 1.
>
> Top 3 explanations of test sample with ID 5 (the numbers in the table are the IDs of the explanation training samples):
>
> |        |  $lr=1e-5$ | $lr=1e-3$ | $lr=0.1$  | $lr=1$ |
> |--------|------------|-----------|-----------|--------|
> | Rank 1 | 13580      | 13580     | 13580     | 7838   |
> | Rank 2 | 7838       | 7838      | 7838      | 13580  |
> | Rank 3 | 7322       | 7322      | 7322      | 7322   |
>
> **If we consider two images from the same class with similar appearances, do they lead to the same explanations?**
>
> Thanks for bringing up such a good question! From our best understanding, it depends on the similarity measure that you are referring to. For example, sky and sea can appear very similar in terms of main colors but they belong to different classes and would have very different explanations.

---

> > ### Author Response · Authors · 2021-08-10
> > **response to reviewer DC2m [2/2]**
> >
> > **It seems that a natural baseline to the data debugging task would be to use the entropy at the output of a model to decide which data points to correct.**
> >
> > Thanks for this comment. Indeed, entropy would be a natural baseline. As shown in the experiment in the Influence Function paper(referred as "loss"), the Influence Function method has already significantly outperformed this baseline. Hence, we didn’t include this weak baseline in our comparison to minimize clutter.  We could add this comparison in an appendix if the reviewer feels it is important to include.
> >
> > **Some experimental details are missing e.g.: which ResNet version is used? How is the RPS lambda chosen when making comparisons? How is the learning rate required by the proposed approach chosen?**
> >
> > Thanks for the comment. We use a ResNet-20. For the $\lambda$ parameter of RPS-$l_2$, we observed in our experiment that the Pearson correlation between the true prediction and the decomposition is quite robust to $\lambda$. Here, we show the effect of tuning $\lambda$ below:
> >
> > The Pearson correlation with respect to different $\lambda$ values on ResNet(round down to keep 3 significant digits)
> >
> > |        | $\lambda=0.1$ | $\lambda=1$ | $\lambda=10$ | $\lambda=100$ |
> > |--------|---------------|-------------|--------------|---------------|
> > | Rank 1 | 3749          | 3749        | 3749         | 3749          |
> > | Rank 2 | 4681          | 4681        | 4681         | 4681          |
> > | Rank 3 | 4947          | 4947        | 4947         | 3154          |
> >
> >
> > Therefore, we choose the $\lambda$ parameter within the common range of the $l_2$ regularization coefficient from $1e-4$ to $3e-3$(also the default $\lambda$ value from the RPS-$l_2$ public codebase).  We will include this discussion in our revision.
> >
> > **The paper does not motivate why explainability approaches which rely on understanding the influence of the training data on the test predictions are important. The paper does not motivate why explaining the predictions of a model is important.**
> >
> > Thanks for the comment. Our motivation is to address the observed problems in the existing RPS-$l_2$. We do acknowledge the importance of explanations for classifiers, as we described in the introduction(line 21 to 25).
> >
> > **It is not clear why diversity in the explanations would be a desirable feature to improve the trustworthiness of a system. Could the authors elaborate on this?**
> >
> > Thanks for the comment. Indeed, diversity is not equivalent to better explanations. Generally speaking, to some extent, the diversity of explanations can provide a more comprehensive understanding of the model.  In this particular context, we aim to address the issue of RPS-$l_2$ where it generates nearly identical explanation samples for the test data in the same class. The diversity is an outcome of our novel derivation of RPS-LJE. It has the benefit of providing instance-level explanations.
> >
> > **Could the authors comment on the use cases of the proposed approach? If the system fails, what happens? How can we mitigate this? What conclusions can we draw about the trustworthiness of the models using the proposed approach? What do we learn from those models that we couldn't learn from previous art?**
> >
> > These are all excellent questions, but we believe many of these questions are an important discussion for the explainable AI research field in general (e.g., trustworthiness) and beyond the scope of what we can definitively address in the present paper. We believe our specific contribution in this paper is to provide an alternative method for model explanations to help human validators inspect model errors. As two key contributions over prior work that we emphasize in the paper, we remark that our model is more lightweight (and efficient) compared to Influence Functions, and provides explanations that are more faithful to the originally trained model in comparison to RPS-$l_2$.
> >
> > ___Clarity___
> >
> > **Figure 1: Include values of l2 weight. Does the observation about diversity hold for different l2 weights?**
> >
> > Thanks for the suggestion. We will include the $l_2$ weights in the next version(from left to right: $[1e-5, 1e-3,1e-2,1e-1]$).The diversity issue preserves. The underlying reason is that the $\alpha_i \phi(x_i)$ dominates the importance calculation. To verify this, we provide a demonstration here comparing the top explanation between different $\lambda$ values on ResNet with CIFAR-10. The results are as following(the numbers in the table are the IDs of the explanation training samples):
> >
> >
> > |                  | Test sample 3 | Test sample  8 | Test sample  12 | Test sample 16 |
> > |------------------|---------------|----------------|-----------------|----------------|
> > | $\lambda =0.001$ | 1499          | 1499           | 1499            | 1499           |
> > | $\lambda =0.01$  | 4275          | 4275           | 4275            | 4275           |
> > | $\lambda =0.1$   | 4275          | 4275           | 4275            | 4275           |
> > | $\lambda =1$     | 3112          | 3112           | 3112            | 3112           |
> > | $\lambda =10$    | 610           | 610            | 610             | 610            |
> >
> >
> > As shown in the table, the issue preserves for a large range of $\lambda$.
> >
> > **Figure 2: Include details such as model used.**
> >
> > Thanks for the suggestion. The model used for CIFAR-10 is ResNet-20. We will add the information in the caption.
> >
> > **Figure 3(b): Include choices of learning rates.**
> >
> > Thanks for the suggestion. The learning rates are stated in the paper (line 177). But, we will highlight this information in the figure caption.
> >
> > **Figure 4 contains a lot of plots. When discussing the figure, it would be good to point the reader towards each specific subplot from which conclusions are being drawn.**
> >
> > Thanks for the suggestion. Most of the discussion is with respect to three plots at the same time. We have one sentence of the individual discussion for the image classification task and will refer to the subplot in the next version.
> >
> > **Figure 5: It could be interesting to share associated importance values.**
> >
> > Thanks for the suggestion. We treat the importance as a ranking problem instead of a classification problem. Therefore, the exact importance value is not the focus.
> >
> > **The model notation introduced in Section 3 seems unnecessary.**
> >
> > Thanks for the suggestion. We introduced the notation $M_{\theta}(x)$ for the classification model so we could refer to the classifier as a function of the inputs in the later line 101 and 111.
> >
> > **Introduce n (line 58).**
> >
> > Thanks for the suggestion. It denotes the number of training examples. We will add the description in the next version.
> >
> > **What is CNN in line 189 (the ResNet model?), same with RNN (the previously introduced LSTM?), it would be better if the authors could stick to the introduced names.**
> >
> > Thanks for the suggestion. We will stick to ResNet and LSTM in the next version.
> >
> > **Although intuitive, random baseline should be explained.**
> >
> > Thanks for the suggestion. We will make the change in the next version.
> >
> > **Appendix is rather packed, it would be helpful to refer to different relevant sections of the appendix in the main body of the paper.**
> >
> > Thanks for the suggestion. We will make the change in the next version.
> >
> > ___Checklist___
> >
> > **The code is not shared. Please revisit 3(a) and 4(c) and indicate that the code will be shared.**
> >
> > Thanks for the comment. We do have plans of releasing the code base (in Pytorch). We will reveal the Github repository link to public access in the final version of the paper.
> >
> > **Please review 4(b) and 4(d). Even if datasets are publicly available, they might come with their licenses, and unfortunately it is not always the case that consent was gathered from people.**
> >
> > Thanks for the comment. We thoroughly checked for CIFAR-10, IMDB sentiment, and German Credit datasets and found that they are all open for academic research purposes.
> >
> > ___Limitation & societal impact___
> >
> > **The reviewer would strongly encourage the authors to move the broader impact to the main body of the paper. The reviewers would strongly encourage the authors to include a limitations discussion, also in the main body of the paper. This is crucial information that should not be disregarded.**
> >
> > Thanks for this suggestion. We acknowledge the importance of these discussions and will move them into the main body of the paper in the next version. For the limitations, we followed the NeurIPS 2021 Paper Checklist Guidelines and reflected on assumptions and the scope of applications as Appendix A.
> >
> > **What are the limitations of the proposed approach? It would seem that one of the limitations is that it would require storing all training data. Could the authors comment on that? Could we consider storing a subset of the training data that would be sufficient to provide explanations? How could we select the data points to consider?**
> >
> > Thanks for bringing up these great questions. As for limitation, we stated our assumption in appendix A that the given model is well-trained and around a saddle point -- we can move this to the main paper. We also discussed the scope of our application: our method can generate explanations for classifiers with a linear output layer. Currently, our method does require storing all training data to generate explanations. We acknowledge that the storage can be a concern with large or sensitive datasets. A possible approach would be to store the data with high gradients or self-influence. However, this is an interesting future direction worth more investigation for the explanation methods that estimate the influence of training data.
> >
> >
> > We hope our response addresses your concerns. If you have any further questions, we would be happy to answer them in follow-up discussion.

---

### Official Review · Reviewer_M1Vv · 2021-07-17

**Rating:** 5
**Confidence:** 5

**Summary:**

The paper proposes a new representer decomposition based on the approximate optimal condition. This address issues for RPS-L2 by 1. this does not rely on regularization strength 2. this provides an explanation that varies much more than RPS-L2 across different testing points. They verify this in their experiments.

**Limitations And Societal Impact:**

yes

**Main Review:**

The paper aims to address issues of RPS-l2, but the key point of RPS-l2 is that the prediction of the test point can be decomposed by training point contribution. The same cannot be said for RPS-LJE as (7) is often not realistic in practice. (7) not realistic will result in (11) being not accurate. In RPS-l2, the prediction of the test point and the sum of training point contribution has >0.95 correlation. I suspect RPS_LJE will get a correlation much lower. Therefore, I view this as a trade-off between the accuracy of the decomposition and the difference between original model and fine-tuned model. Moreover, the issues of RPS-l2 can be alleviated by changing the lambda parameter, as when lambda is large, alpha_i will not dominate. When lambda is small, the difference between original model and fine-tuned model will be small. When you don't fine-tune the model until exact convergence with large lambda, then both issues may go away (with the additional downside that the decomposition is not accurate). However, the proposed RPS_LJE is also not accurate in decomposition. Without a better empirical study of the trade-off for RPS_LJE and RPS_l2 with different strategies, the advantage of RPS_LJE is unclear.

However, the main issue is that the method seems in spirit the same as the idea of influence function. I am confident that if you replace the  theta^* by original theta in (11), you will get almost the same explanation result (this should be a control experiment). After the replacement, (11) and (13) are almost identical. Since the inverse hessian term is the same, and thus the calculation cost of influence function carries over. Therefore, I cannot imagine any use cases where RPS_LJE will be chosen over the influence function. Based on this, I really feel the paper has limited novelty and contribution. (in fig 4, one image out of 16 images of proposed method and influence function is different).

To improve the quality of the paper, the authors should 1. evaluate how well F(x) correlates with \sum_i ai K(x,x_i) 2. compare RPS_LJE with modification of RPS_L2 with different alpha and different optimization strategy 3. point out use cases where RPS_LJE may be more useful than the influence function.

From a practical viewpoint, point 3 is especially important. If the authors cannot do the third point, the paper would be better as a decomposition viewpoint of the influence function (if point 1 is achieved), but not a new method. If the authors can provide more (convincing) use cases in practice when RPS-LJE is better than the influence function, I will consider raising my score.

**Time Spent Reviewing:**

3

---

> ### Author Response · Authors · 2021-08-10
> **response to reviewer M1Vv**
>
>
> Thanks for your feedback. We are happy to respond to your comments and questions as below:
>
> **Q1: (7) is often not realistic in practice.  (7) not realistic will result in (11) being not accurate.**
>
> Thanks for this valuable concern. The RPS-$l_2$ leverages a fine-tuning step to avoid the assumption. In our case, we assume the given model is trained to a local optima and near a saddle point, where the first order derivative is approximately zero. The convergence to a local optima is also the general stopping criteria for neural network training. Similarly, the Influence function [7] work also makes this assumption in equation 10. However, early stopping is also widely adopted stopping criteria in neural network training in practice and may violate this assumption. We do agree that the performance of our methodology will be affected by the violations. In the appendix section A, we provide a discussion of this assumption. While RPS-$l_2$ leverages the fine-tuning step to avoid this assumption, it makes an implicit assumption that the fine-tuned model is similar to the original model. However, the fine-tuning step with the added $l_2$ term can completely alter the original model. As our experiment demonstrates(Figure 1), the effect that different $l_2$ coefficients result in different explanations, making the interpretations questionable for the original model.
>
> **Q2: In RPS-$l_2$, the prediction of the test point and the sum of training point contribution has >0.95 correlation. I suspect RPS_LJE will get a correlation much lower. The author should evaluate how well F(x) correlates with $\sum_i \alpha_i K(x,x_i)$.**
>
> Thanks for this valuable concern. To verify the correlations between the true prediction and the decomposition, we conducted an experiment to compare the Pearson correlation on ResNet as the RPS-$l_2$ does for both training and testing data. We further report the correlation on LSTM.
>
> Here is the result for training data(round down to keep two significant digits):
>
>
> | Method  	| ResNet | LSTM  |
> |-----------|--------|-------|
> | RPS-$l_2$ | 0.99   | 0. 99 |
> | RPS-LJE 	| 0.99   | 0. 99 |
>
> Here is the result for testing data(round down to keep two significant digits):
>
> | Method    | ResNet | LSTM |
> |-----------|--------|------|
> | RPS-$l_2$ | 0.99   | 0.99 |
> | RPS-LJE   | 0.99   | 0.98 |
>
>
> All correlation values are close to 1, thus showing that for RPS-LJE, $F(x)$( $\hat{\mathbf{y}}_t$ in the paper) also highly correlates with $\sum_i \alpha_i K(x,x_i)$ .
>
>
> **Q3:The issues of RPS-$l_2$ can be alleviated by changing the lambda parameter, as when lambda is large, $\alpha_i$ will not dominate.**
>
> Thanks for this comment. As shown in equation 5, the underlying reason for the similarity between explanations for the same class test points is because the term $\alpha_i \phi(x_i) = -\frac{1}{2\lambda n}\frac{\partial \mathcal{L}(\mathbf{x}_i, y_i, \boldsymbol{\Theta^\*})}{\partial \Theta_L^\*}$ dominates. Therefore, even for large $\lambda$, the term still appears to dominate the calculation.
>
> To verify this, we provide a demonstration below that compares the top explanation between different $\lambda$ values on ResNet with CIFAR-10. The results are as follows (the numbers in the table are the IDs of the explanation training samples):
>
>
> |                  | Test sample 3 | Test sample  8 | Test sample  12 | Test sample 16 |
> |------------------|---------------|----------------|-----------------|----------------|
> | $\lambda =0.001$ | 1499          | 1499           | 1499            | 1499           |
> | $\lambda =0.01$  | 4275          | 4275           | 4275            | 4275           |
> | $\lambda =0.1$   | 4275          | 4275           | 4275            | 4275           |
> | $\lambda =1$     | 3112          | 3112           | 3112            | 3112           |
> | $\lambda =10$    | 610           | 610            | 610             | 610            |
>
>
> Even with large $\lambda$ values, the explanations are identical for the test sample within the same class.
>
> **Q4: When lambda is small, the difference between original model and fine-tuned model will be small.**
>
> Thanks for this comment. Even when $\lambda$ is small, the model at equilibrium after the fine-tuning step is not necessarily close to the original model. The additional $l_2$ term makes the loss function a different optimization objective.
>
> **Q5:The author should compare RPS_LJE with modification of RPS_L2 with different alpha and different optimization strategy.**
>
> Thanks for the valuable suggestion. As shown above, we provided additional result tables demonstrating 1) the high correlation between the true prediction and decomposition for both ResNet and LSTM. 2) large $\lambda$ values do not solve the problem of class-level explanation.
>
> **Q6:The main issue is that the method seems in spirit the same as the idea of influence function. The author should point out use cases where RPS_LJE may be more useful than the influence function.**
>
> Thanks for your valuable concern. Our aim of this paper is to address the problems we observed in the existing RPS-$l_2$ method. We theoretically overcome the two issues with our novel derivation of our proposed RPS-LJE method. We acknowledge that the resulting objective in equation 11 is quite close to Influence Function but with two differences, as discussed in section 3.3:
> 1. The Influence Function includes a multiplicative factor $\frac{\partial \mathcal{L}(\boldsymbol{x}_t,y_t,\boldsymbol{\Theta}^\dagger)}{\partial \Theta_L^\dagger\phi(\boldsymbol{x}_t)}$ that is relevant to the test point, whereas the RPS-LJE has an additive factor $\frac{1}{n}\Theta_L^*\phi(\boldsymbol{x}_t)$ that is also only relevant to the test point.
> 2. The derivative in the Influence Function respects the original model parameters $\Theta^\dagger$, whereas RPS-LJE's derivatives apply to $\Theta^*$.
>
> While the results look similar, the derivations are very different. We derive our result from the RPS perspective, instead of the Influence Function perspective.
>
> Another key difference between the RPS methods (including RPS-$l_2$ and RPS-LJE) and the influence function is that the RPS methods only work with the parameters in the last linear layer instead of the whole neural network. When efficiency is a concern, we would suggest users to choose RPS-LJE over the Influence Function for this reason.
>
> Lastly, we hope that our methodology provides another alternative to users for model explanations and that our formalization and derivation can serve as a (currently missing) link to connect the existing influence function and RPS-$l_2$ methodologies.
>
>
> We hope our response addresses your concerns. If you have any further questions, we would be happy to answer them in follow-up discussion.

---

> > ### Comment · Reviewer_M1Vv · 2021-08-11
> > **Thanks for the response**
> >
> > Influence function are operated on the last linear layer in practice also, so I do not buy the efficiency statement. I would view the contribution of the paper as providing a representer decomposition method for the influence function. However, in this case, you need to connect the influence function and RPS-LJE.
> >
> > I suggest the authors to better understand why the result of the RPS-LJE is so similar to the influence function. By better understanding the similarity, you may also better understand the difference between them. A first step is to look at the Pearson and Spearman correlation for RPS-LJE and influence function for some test points. If the correlation is not high, I would investigate which example has the most difference between the two methods. Does removing/ add some terms connect the two methods.

---

> > > ### Author Response · Authors · 2021-08-13
> > > **response to reviewer M1Vv**
> > >
> > >
> > > Thanks for your comment. We agree that the Influence Function methodology would be much more efficient when restricted to the final layer; however, the formal derivation of Influence Function requires calculating gradients with respect to all parameters while RPS methods (RPS-LJE and RPS-$l_2$) only focuses the last layer. We are aware the final expression appears similar to the one used in the influence function paper, which is the reason why we wrote Section 3.3 to clarify the similarities and differences. We would also like to reiterate that from a technical perspective, our derivation connects the RPS and Influence Function methodologies, which we believe is an important and novel contribution since previously these were considered two distinct and different methods.
> > >
> > > Thanks for the brilliant experimental suggestions. Based on your suggestion, we conducted the following experiments where we calculate the Pearson correlations between the importance scores and calculated by RPS-LJE and Influence Functions for the top-5 explanation training points (we take the union of the top-5 explanations from the two methods). We also calculate the Spearman correlations between the rankings of the same group of explanations. The results are shown below (we report the distributions in quantiles):
> > >
> > > Pearson correlations
> > >
> > > | Quantile of distribution | 0.01%   | 0.1%    | 1%      | 10%     | 30%     | 50%     | 70%    | 90%    |
> > > |--------------------------|---------|---------|---------|---------|---------|---------|--------|--------|
> > > | ResNet                   | 0.9998  | 0.9998  | 0.9999  | 0.9999  | 0.9999  | 0.9999  | 0.9999 | 0.9999 |
> > > | LSTM                     | 0.1484  | 0.7788  | 0.9513  | 0.8855  | 0.9992  | 0.9997  | 0.9999 | 0.9999 |
> > > | Xgboost                  | -0.8628 | -0.8628 | -0.7201 | -0.5694 | -0.2998 | -0.0569 | 0.2451 | 0.6025 |
> > >
> > >
> > > Spearman correlations
> > >
> > > | Quantile of distribution | 0.01%   | 0.1%    | 1%      | 10%     | 30%     | 50%     | 70%    | 90%    |
> > > |--------------------------|---------|---------|---------|---------|---------|---------|--------|--------|
> > > | ResNet                   | 0.7000  | 0.9000  | 0.9428  | 1.0000  | 1.0000  | 1.0000  | 1.0000 | 1.0000 |
> > > | LSTM                     | 0.1000  | 0.4000  | 0.7000  | 0.9000  | 1.0000  | 1.0000  | 1.0000 | 1.0000 |
> > > | Xgboost                  | -0.8214 | -0.8214 | -0.8181 | -0.6666 | -0.3833 | -0.1428 | 0.0714 | 0.4285 |
> > >
> > >
> > > Our results show, while in many cases the two approaches provide similar explanations, there are cases where explanations are completely different. This observation reflects the fundamental difference between equation (11) and (13), where equation (11) includes an additional factor and equation (13) includes a multiplicative factor. This experiment demonstrates that even though the two approaches appear similar, the subtle difference still makes the two approaches show different behaviour, even in practice.

---

> > > > ### Comment · Reviewer_M1Vv · 2021-08-24
> > > > **How do you interpret the result?**
> > > >
> > > > I see the numbers, but I have no idea why the numbers are so positive in Resnet and lower in LSTM and Xgboost. I also don't see the authors providing insights. Most importantly, when they are different why is RPS-LJE better? From Xgboost examples I cannot judge the quality. Without such insights and understanding of the difference, when should one consider RPS-LJE over the influence function? When will they be almost exactly the same?
> > > >
> > > > In theory, the influence function can be applied on the last layer by assuming only the last layer is trained without the data point. This is the same assumption made implicitly by RPS-L2 and RPS-LJE. I don't see the last layer thing as an advantage.
> > > >
> > > > If your correlation is 1, I can interpret RPS-LJE as decomposition of the prediction over training examples (with satisfies the completeness axiom for data importance method). Since they are different, I cannot view RPS-LJE as a bridge between the methodology of RPS-LJE and influence function. I still don't see when RPS-LJE will be used over the influence function.

---

### Official Review · Reviewer_Tvmz · 2021-07-17

**Rating:** 7
**Confidence:** 4

**Summary:**

This paper addresses sample based explainable AI, where a model prediction at test time is explained by producing a ranked list of training set instances.  They build upon a method called Representer Point Selection, which develops a representer theorem wherein contribution to test loss is decomposed into terms involving the features of the last layer the trained network.  The authors address two shortcomings of the method by developing  a local Jacobean Taylor expansion that alleviates both.  They present experiments on data set cleaning (in line with the original RPS paper), as well as on three other tasks commonly evaluated with sample based explainability.

**Ethical Concerns:**

None.

**Limitations And Societal Impact:**

No code provided, but no other relevant comments here.

**Main Review:**

The authors take a narrow view of explainable AI for error analysis through dataset inspection.  However, they present their case for representer point selection through local Jacobian expansion (RPS-LJE) quite clearly.  They address the following two deficiencies of RPS.  First, RPS as originally proposed requires fine-tuning of the last layer network weights with $\ell_2$ regularization optimized via L-BFGS to drive the magnitude of the gradient to vanishing, creating a discrepancy between the original model making predictions, and the resultant model providing a link between training points and test predictions.  Second, they provide evidence that most high ranked training set samples are identical for test samples sharing the same class, which limits its utility for explaining individual predictions.

Their main technical contribution is in reformulating the model prediction $\hat{\mathbf{y}_t} = \Theta\_{L}^{m}\phi{\mathbf{x}_t}$ so that it is linear in the contribution of the training points but without introducing $\lambda$ or leaving it dominated by the contribution of the gradient of the loss wrt the modified parameter set, laid out in sections 3.1, 3.2 and especially equation 11.

A few comments:

1. Equation 11 contains the inverse of a Hessian matrix evaluated at a nearby set of parameters $\Theta^{*}_{L}$.  Even for small sets of parameters, the Hessian can be difficult to estimate and may require the addition of a damping term on the diagonal to ensure it is positive definite (the original Influence Functions paper by Koh and Liang encounter this).  Did the authors encounter any practical necessity to ensure the Hessian is positive definite?

2.  The qualitative experiments in section 4.3 are somewhat underwhelming.  It is not clear by my reading why the highlighted examples in either Credit Risk analysis or sentiment analysis with LSTM convince me of the method's validity.  They do establish that the method can work on models beyond convolutional (or feed forward) networks, but perhaps a more convincing set of experiments with a more relatable outcome would help?  For example, in the case of sentiment analysis, this could take the form of generating adversarially perturbed training data to augment the training set, and plotting the rank distribution of the adversarially altered negative control training data against the top ranked bona fide training data.

Finally, this paper would be more immediately useful if it released code to implement RPS-LJE in either TensorFlow or PyTorch (as the original RPS paper did).

Overall, I found this a very clearly written paper that presented a solid if narrow contribution, and should be a clear accept.



**Time Spent Reviewing:**

5

---

> ### Author Response · Authors · 2021-08-10
> **response to reviewer Tvmz**
>
> Thanks for your feedback. We are happy to respond to your comments and questions as below:
>
> **Q1: Equation 11 contains the inverse of a Hessian matrix evaluated at a nearby set of parameters $Θ_L^∗$. Even for small sets of parameters, the Hessian can be difficult to estimate and may require the addition of a damping term on the diagonal to ensure it is positive definite (the original Influence Functions paper by Koh and Liang encounter this). Did the authors encounter any practical necessity to ensure the Hessian is positive definite?**
>
> Thanks for your comment. Indeed, even with the small set of parameters in the last linear layer, the Hessian matrix calculation can be complicated and may not be positive definite. In the particular experiments in our paper (Section 4.3), we focus on binary classification, where the binary cross entropy loss is strictly convex when the features span the space.  Because these conditions held in our experiments, our Hessian computations always yielded a positive definite result.  However, in the general case, the Hessian matrix may not be positive definite as the cross entropy loss is convex but not always strictly convex.  We will add in the positive definite assumption in the next version. One potential solution is to adopt the method used in the Influence Function paper with the damping term.
>
> **Q2:The qualitative experiments in section 4.3 are somewhat underwhelming. It is not clear by my reading why the highlighted examples in either Credit Risk analysis or sentiment analysis with LSTM convince me of the method's validity. They do establish that the method can work on models beyond convolutional (or feed forward) networks, but perhaps a more convincing set of experiments with a more relatable outcome would help? For example, in the case of sentiment analysis, this could take the form of generating adversarially perturbed training data to augment the training set, and plotting the rank distribution of the adversarially altered negative control training data against the top ranked bona fide training data.**
>
> Thanks for this brilliant suggestion. As described in Section 4.3, we showed that RPS-LJE explanations are more coherent to the test points. (For LSTM, the sentence format and key words of the explanations are more similar to the test points. For Xgboost, the features are more similar.) Furthermore, we do acknowledge the advice of using adversarial examples to support our claim. Therefore, we added the experiment below where we perturb a training example by substituting a key word with its synonym and observe the rank 1 explanations generated by the three methods:
>
>
> |                    | id | review text                                                           |
> |--------------------|-------------|-----------------------------------------------------------------------|
> | Original sample    | 2619        | Wow, this was another good spin off of the original American pie…     |
> | Perturbed sample   | 2619        | Wow, this was another *great* spin off of the original American pie…    |
> | Influence Function | 2619        | Wow, this was another good spin off of the original American pie…     |
> | RPS-$l_2$          | 14701       | This is a very memorable spaghetti western. It has a great storyline… |
> | RPS-LJE            | 2619        | Wow, this was another good spin off of the original American pie..    |
>
>
> |                    | **id**    | **review text**                                                                     |
> |--------------------|-------|---------------------------------------------------------------------------------|
> | Original sample    | 4789  | Simply the best Estonian film that I have ever seen, although it is...     |
> | Perturbed sample   | 4789  | Simply the *greatest* Estonian film that I have ever seen, although it is... |
> | Influence Function | 4789  | Simply the best Estonian film that I have ever seen, although it is...     |
> | RPS-$l_2$          | 14701 | This is a very memorable spaghetti western . It has a great storyline...        |
> | RPS-LJE            | 4789  | Simply the best Estonian film that I have ever seen, although it is...     |
>
>
>
> |                    | **id**    | **review text**                                                                  |
> |--------------------|-------|------------------------------------------------------------------------------|
> | Original sample    | 11177 | I can't tell you all how much I love this movie. I have read reviews... |
> | Perturbed sample   | 11177 | I can't tell you all how much I *like* this movie. I have read reviews... |
> | Influence Function | 11177 | I can't tell you all how much I love this movie. I have read reviews... |
> | RPS-$l_2$          | 9112  | Tim Krabbe is the praised author of ' Het Gouden Ei ' , a novel that...      |
> | RPS-LJE            | 11177 | I can't tell you all how much I love this movie. I have read reviews... |.
>
>
> As shown in the tables, both Influence Function and RPS-LJE are able to rank the original training sample as the top explanation. While Influence Function and RPS-LJE perform comparably in this case, as demonstrated in section 4.3,  we observe that RPS-LJE yields training data explanations that are more similar to the test examples in comparison to Influence Function.
>
> **Q3: This paper would be more immediately useful if it released code to implement RPS-LJE in either TensorFlow or PyTorch (as the original RPS paper did).**
>
> Thanks for your suggestion. We do have plans of releasing the code base (in Pytorch). We will reveal the Github repository link to public access in the final version of the paper.
>
> We hope our response and additional experiments address your concerns. If you have any further questions, we would be happy to answer them in follow-up discussion.

---

> > ### Comment · Reviewer_Tvmz · 2021-08-24
> > **Thanks for your thoughtful rebuttal contributions**
> >
> > > In the particular experiments in our paper (Section 4.3), we focus on binary classification, where the binary cross entropy loss is strictly convex when the features span the space. Because these conditions held in our experiments, our Hessian computations always yielded a positive definite result. However, in the general case, the Hessian matrix may not be positive definite as the cross entropy loss is convex but not always strictly convex. We will add in the positive definite assumption in the next version. One potential solution is to adopt the method used in the Influence Function paper with the damping term.
> >
> > I think this remark will really help people in practice.  It reduces the possibility of the pitfall of applying RPS-LJE in a situation where $H$ would not be positive definite, and provides a sensible remedy.  Perhaps the remark could also include a suggestion to check the eigenvalues of $H$, and then suggest the damping term remedy (cf. the IF paper).
> >
> > > Therefore, we added the experiment below where we perturb a training example by substituting a key word with its synonym and observe the rank 1 explanations generated by the three methods:
> >
> > Thanks for including this experiment.  It isn't quite what I had intended, but it's helpful.  I was imagining both a synonym *and* antonym experiment where you would demonstrate that (a) the rank of synonymous substitutions is largely unchanged, and (b) the rank of antonyms is much lower.

---

### Decision · Program_Chairs · 2021-09-27

**Decision:**

Accept (Poster)

**Comment:**

This paper proposes a variant of RPS called RPS-LJE that is claimed to offer several advantages over the previously proposed RPS-L2. The “explanation” acts on the last layer of deep nets only, atributing predictions to training points while treating the neural network as fixed. As such, it seems unreasonable to call this (or any related works that act similarly) explanatinos of “deep neural networks” vs of linear models wiith some fixed basis. The conversation centered around several claimed advantages — efficiency (it appears to be resolved in discussion that there is no efficiency gain over influence functions), a faithfulness advantage over RPS-L2, and the striking similarity and high agreement with influence function, raising the question of whether the method is really adding anything or if the novelty is all in the new perspective/derivation.

The rebuttal and discussion were thoughtful and all reviewers were engaged. However, while they converged on several points of fact, a few others were left unresolved and their were some genuine normative disagreements about whether the contributions that withstood the test of the debate were sufficient to warrant acceptance.

I will not hide the fact that this is a difficult metareview to write and that I am not sure how to resolve the remaining impasse. I am still seeking additional opinions and hope to make a more confident assessment but believe this is a genuinely difficult case. My current best assessment now is that this paper is truly borderline. However, I believe that with reasonable adjustments to claims and presentation per promises made in the discussion period, this paper is honest work and we can leave it to posterity (vs the review process) to determine the work's eventual impact.